# MARS an improved de novo peptide candidate selection method for non-canonical antigen target discovery in cancer

Hanqing Liao[1,2], Carolina Barra [3], Zhicheng Zhou [4], Xu Peng[1], Isaac Woodhouse [1,2], Arun Tailor [1,2], Robert Parker[1,2], Alexia Carré [4], Persephone Borrow [2], Michael J. Hogan [5], Wayne Paes [1,2], Laurence C. Eisenlohr [5,6], Roberto Mallone [4,7], Morten Nielsen [3,9] & Nicola Ternette [1,2,8,9] ✉

Understanding the nature and extent of non-canonical human leukocyte antigen (HLA) presentation in tumour cells is a priority for target antigen discovery for the development of next generation immunotherapies in cancer. We here employ a de novo mass spectrometric sequencing approach with a refined, MHC-centric analysis strategy to detect non-canonical MHC-associated peptides specific to cancer without any prior knowledge of the target sequence from genomic or RNA sequencing data. Our strategy integrates **M**HC binding rank, **A**verage local confidence scores, and peptide **R**etention time prediction for improved de novo candidate **S**election; culminating in the machine learning model MARS. We benchmark our model on a large synthetic peptide library dataset and reanalysis of a published dataset of high-quality non-canonical MHC-associated peptide identifications in human cancer. We achieve almost 2-fold improvement for high quality spectral assignments in comparison to de novo sequencing alone with an estimated accuracy of above 85.7% when integrated with a stepwise peptide sequence mapping strategy. Finally, we utilize MARS to detect and validate lncRNA-derived peptides in human cervical tumour resections, demonstrating its suitability to discover novel, immunogenic, non-canonical peptide sequences in primary tumour tissue.

The analysis of MHC-bound peptide ligands using liquid chromatography-mass spectrometry (LC-MS) technology (immuno-peptidomics) is a rapidly evolving field that has advanced identification of T cell antigens, including clinically relevant epitopes in autoimmunity, pathogen infection and cancer[1–3].

Software tools to interrogate LC-MS spectral data have been widely adopted from the field of proteomics, and extensive differences in performance have been observed for MHC-derived peptide data[4–6]. Spectral interpretation software is based on three main strategies. The most common approach, the "database search" approach[7–9], relies

[1]The Jenner Institute, University of Oxford, Oxford OX3 7BN, UK. [2]Centre for Immuno-Oncology, Nuffield Department of Medicine, University of Oxford, Oxford OX3 7DQ, UK. [3]Technical University Denmark, Copenhagen, Denmark. [4]Université Paris Cité, Institut Cochin, CNRS, INSERM, 75014 Paris, France. [5]Department of Pathology and Laboratory Medicine, Children's Hospital of Philadelphia, Philadelphia, PA 19104, USA. [6]Department of Pathology and Laboratory Medicine, Perelman School of Medicine, University of Pennsylvania, Philadelphia, PA 19104, USA. [7]Assistance Publique Hôpitaux de Paris, Service de Diabétologie et Immunologie Clinique, Cochin Hospital, 75014 Paris, France. [8]University of Utrecht, Department of Pharmaceutical Sciences, 3584 CH Utrecht, The Netherlands. [9]These authors contributed equally: Morten Nielsen, Nicola Ternette. ✉e-mail: nicola.ternette@ndm.ox.ac.uk

on matching the masses of experimentally obtained peptide fragment spectra to all theoretically possible peptide fragments of sequences in a protein sequence list or "database". This approach is limited to identifications in the sequence database and can therefore not be used for the discovery of non-canonical peptides unless such information is added to known canonical protein databases. This approach has further been extended by integration of spectral matching assignment to increase sensitivity, which uses advanced deep learning and peptide fragmentation prediction to annotate the experimentally obtained peptide fragment spectra with high accuracy[10–14].

Thirdly, de novo algorithms interpret the MS spectra without comparison to a protein sequence database, annotating the amino acid (aa) sequence directly from the spectral information available[15–17]. Integration of de novo approaches and database searching (de novo-assisted database approach) has been shown to be particularly advantageous for MHC-peptide datasets[4,5,18].

For the interrogation of immunopeptidome datasets with a focus on cancer-specific peptide ligands, proteogenomics approaches have been used to refine the database search approach in order to identify peptide sequences that contain tumour-specific mutations or other non-canonical sources like functional RNA genes, intronic sequences, and/or reading frames in unannotated genomic origins that are generally not present in universal protein databases[6,18–23]. Here, genomics and/or transcriptomics data is integrated to achieve a highly accurate, sample-specific database that can be used as template for a database search. These approaches have recently been expanded to identification of peptides from unannotated open reading frames (ORFs), or out-of-frame translated sequences, that are valuable for the development of immunotherapeutic approaches. However, the observation that many MHC-presented peptides exhibit minimal RNA expression[24] highlight the need for transcriptomics data-independent interrogation of the immunopeptidome for cancer antigen discovery.

The purpose of this study was to increase the power of the de novo approach for identification of non-canonical MHC-I-associated peptides without the use of genomics or transcriptomics data and/or personalized template protein databases.

De novo approaches have so far not often been integrated for peptide antigen discovery, due to the fact that peptide fragmentation by MS/MS is never fully "complete", and does frequently lack consecutive ion series, particularly in MS/MS data in which the basic C-terminus and a strong y-ion series is not specifically generated by tryptic digestion. Gaps in the fragment ion series allow alternative sequence interpretation, resulting in identical probability scores for isobaric aa combinations and sequence permutations.

This is specifically relevant for immunopeptidomics experiments: Since the MHC gene locus (in humans called human leukocyte antigen, HLA) is the most polymorphic in the human genome, each individual allele binds peptides with specific aa in the main anchor positions, generally positions 2,3 and the C-terminus. Every person expresses up to six classical MHC-I proteins with such restricted sequence specificities. The resulting peptide repertoires that are purified exhibit the respective sequence clusters formed by the MHC alleles present in each sample. Consequently, the quality of the measured spectra varies for each sequence subcluster, depending on the originating HLA allele sequence restriction.

Hence, we postulated that we could exploit our existing knowledge of the peptide sequence restriction defined by the MHC alleles present in the sample for 'filling the gaps' in incomplete fragment spectra, and ultimately achieve a more accurate sequence assignment. We had previously shown that MHC binding prediction can rescue low confidence identification in database approaches (MSRescue[25]). Recently, MHC binding prediction has been applied in a personalized fashion in order to identify patient neoantigens[26], and for rescoring of database search results for improved spectral identification accuracy[27].

The chromatographic retention time (RT) of a peptide is an independent property of a peptide that could be used to further select amongst sequence candidates. RT prediction based on deep learning is now very accurate. It has been shown that integrating the difference between observed and predicted RT to a database-search-based peptide identification workflows can improve performance (DeepRTplus[28], Deep Rescore[14]).

We report here that by integration of (i) MHC binding affinity predictions and (ii) differences between observed and predicted RT of candidate peptide sequences as additional factors, MHC-I peptide identification sensitivity can be improved over the baseline de novo sequencing annotation. We validate our findings by benchmarking our algorithm termed MARS on LC-MS data from synthetic MHC-I peptide libraries and published non-canonical peptide sequences that were obtained from proteogenomic analyses. We then proceed to develop a hierarchical strategy to map MARS sequence candidates to their likely origin in the genome and obtain a final false discovery rate (FDR) < 14.3%, equating to a sequence assignment accuracy of ~85%. These results offer opportunity for improved shortlisting of accurate sequence identifications from the generally observed global 35% prediction accuracy for complete peptide sequences in de novo sequence data[29]. Through this approach, we are able to expand the non-canonical peptidome of published data by hundreds of peptides. We finally detect and validate long non-coding (lnc) RNA-derived peptides in primary human tumour tissue, demonstrating their immunogenicity and applicability for antigen discovery in cancer.

## Results

### Training data selection and processing

Our aim was to construct and validate our computational model for improved de novo (template-independent) identification of MHC-I-associated peptides. We selected a panel of in-house single HLA allele (CD4.221 cells expressing either A*01:01, A*02:01, A*03:01, A*11:01, B*08:01, B*44:02, B*57:03, C*03:03, or C*03:04) and multi-allele (Jurkat cell line: A*03:01, B*07:02, B35:03, C*04:01, C*07:02; and the C1866 cell line: A*24:02. B*40:02, B*51:01, C*03:03, C*14:02) datasets to develop models and evaluate their performance.

We analyzed these datasets using Peaks DeNovo and also the Peaks DeNovo-assisted Database (DB) search using the SwissProt human canonical proteome[30]. From the Peaks DeNovo search, we obtained up to 100 sequence candidates per spectrum (independent from the provided sequence database), with associated average local confidence (ALC) scores for each candidate sequence as a measure for the confidence of each sequence assignment. The Peaks DB search results were returned as a list of sequence assignments that matched to the SwissProt database. This latter search results in one sequence assignment per spectrum (with exception of L/I isomers) with an associated score (−10$lg$P) reflecting the probability of a correct peptide spectrum match. The results are controlled by a simultaneous decoy DB search, which provides an accurate estimation of the FDR to be expected in the result[31].

To define a subset of "true sequences" to use as a training dataset, we defined spectra as confidently assigned if they were identified with Peaks DB with high confidence −10$lg$P ≥ 20 resulting in a FDR < 1% on peptide-spectrum match (PSM) level. In addition, we only included peptides with a length of 8–13 aa, and those that were assigned a low NetMHCpan 4.1 Eluted Ligand Rank score (M, M ≤ 10), aiming to remove peptides that are likely originating from co-purification during biochemical enrichment. All spectra that fulfilled these criteria were defined as the training dataset.

Next, within the Peaks DeNovo search results, the "true" peptide was considered the Peaks DB assigned sequence identification with all constituent isoleucine (I) replaced by leucine (L), because the here applied MS approach cannot distinguish I from L due to their identical atomic composition and mass. Only those spectra with the "true" (DB)

peptides present in the list of the de novo peptide candidates were selected for model development and evaluation. To confirm that the I to L replacement does not negatively affect the NetMHCpan binding prediction, we tested NetMHCpan predictions within the test set for both L and I variants and found that replacement of all I with L incurred significantly less divergence from the original result than vice versa (Supplementary Fig. 1).

All selected spectra (118,177 spectra in total) were then split into 5 equal parts to form a five-fold cross-validation scheme, such that no two parts shared any common 8-mer substring training peptide sequence. The model trainings and evaluations were carried out using a five-fold cross-validation scheme, where each time a different combination of four parts was used as model-training set, while the performances were evaluated on the fifth (test) partition. Finally, all five-fold test set predictions were concatenated to obtain the final model performance.

### M + A α-model evaluation: incorporation of NetMHCpan prediction improves de novo candidate selection

We used the Peaks DeNovo result output as the baseline model. Accordingly, for each spectrum, the peptide candidate of the highest ALC (A-score) was taken as the "identified" peptide. In order to understand whether we could re-rank the de novo candidates by selecting the best candidate based on their binding prediction to the respective MHC allele of origin, we determined the NetMHCpan binding percentile eluted ligand rank score (M-score) for each of the 100 sequence candidates in the PEAKS DeNovo list and calculated an affine (linear) combination[32] of the A- and M-scores for each peptide candidate to produce a new composite score. Then, for each spectrum, the peptide candidate of the highest composite score was defined as the identified peptide. Because this affine combination was controlled by a single weight parameter α, and modelled A-score and M-score as two peptide identity determining factors, we named it "two-factor (M + A) α model". We calculated the F-ranks which describe the rank of the true peptide sequence in the list of peptide sequence candidates ordered by score, divided by the total number of sequence candidates.

The F-rank therefore provides a normalized performance measure for each spectrum, and the smaller the F-rank scores, the better the sequence candidate scoring performs. We applied a grid search to obtain the optimal α value for a minimal mean F-rank within the training set. Then, this α value was applied to the test for performance evaluation. All five test sets yielded one-sided paired Wilcoxon signed-rank test $p$ values < 2.2E10-16, indicating that the reduction of F-ranks by applying the (M + A) α model in comparison to the ALC baseline model was highly significant (Supplementary Table 1, Fig. 1A, B).

To examine whether the improved F-rank led to higher identification sensitivity when top scoring peptide candidate per spectrum was taken as its identity, we employed another measurement, the full sequence recall (FSR), which depicts the proportion of identified (top one scoring) peptides being correct. A paired t-test using the mean FSRs from each of the five partitions also suggested that the (M + A) α model (M = 0.782, SD = 0.00462) significantly improved the FSR over the baseline A-score model (M = 0.731, SD = 0.00554); t(4) = 36.74, $p$ = 3.28E10-6 (Fig. 1C).

Lastly, the optimal α obtained in the five-fold cross-validations were highly concordant, suggesting the model was robust (Supplementary Table 1). In summary, these results suggested that incorporating MHC binding prediction as a factor can improve PSM candidate selection, resulting in improved peptide identification compared to de novo sequencing alone.

### Incorporation of RT prediction further improves de novo candidate selection: final model evaluation

To further improve PSM candidate selection, we incorporated DeepRTplus[28] to predict the RT of input peptides. We trained the DeepRTplus model with our training data and applied the learned model to the training data itself to derive a score defined through an absolute value of differences between predicted and observed RT (R-score). We first evaluated the integration of A-score and R-score in a two-factor (A + R) β model and confirmed its improved performance over the A-score model alone. While we saw overall improvement of performance, the effect achieved with integrating the R-score resulted

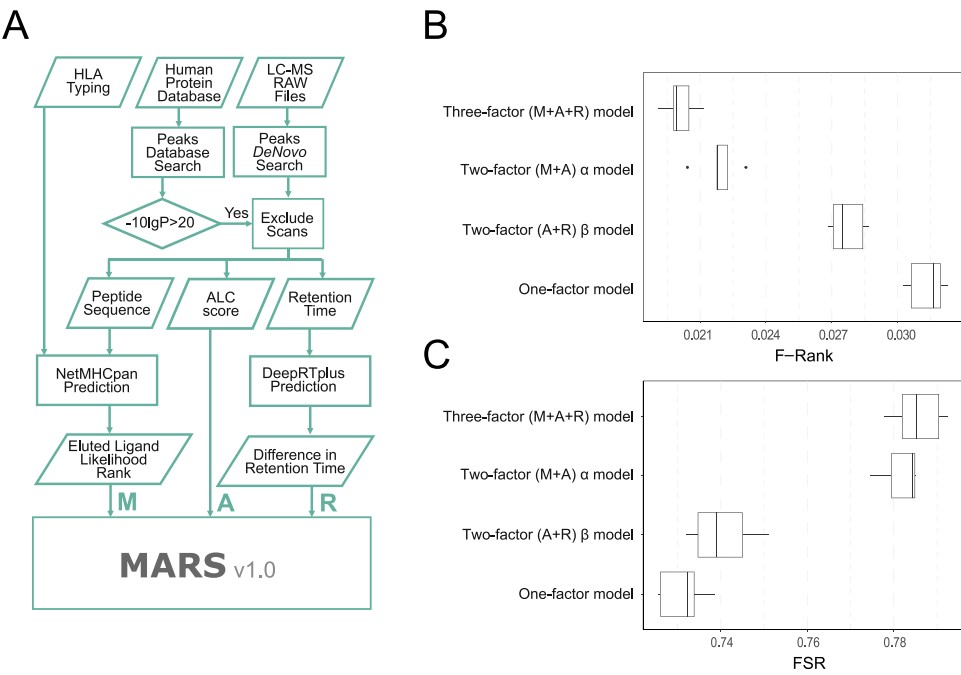

**Fig. 1 | Schematic of MARS workflow and performance evaluation in a five-fold cross-validation. A** Schematic of integrative modules in the MARS pipeline (**B**) Box-plots depicting the F-rank score distributions and (**C**) Full sequence recall (FSR) achieved across the partitions in the target peptide space (118,177 spectra). The boxes define the upper to lower quartiles and the median value is indicated within the box as vertical line, while whiskers indicate the minimum and maximum values, respectively. Outliers are indicated individually as data points. Source data are provided as a supplementary file for Fig. 1B and C.

in only a modest improvement in comparison to the integration of the M-score in the two-factor (M + A) α model (Fig. 1B). We then defined an affine combination of the M-, A- and R-scores to define a three-factor composite model (M + A + R) and used the grid search to find optimal α (weight on M-score) and β (weight on R-score) values to minimize the mean F-rank in the training data (Supplementary Table 1). Finally, the optimal values of α and β were used to assign combinatory scores to each peptide in the test data.

We then proceeded with our evaluation strategy as before: a paired Wilcoxon test revealed that the F-ranks of the (M + A + R) model were significantly lower than those of the (M + A) model (maximum of five p-values < 1.34E-10 (Supplementary Table 1, Fig. 1B), suggesting that the three-factor (M + A + R) model outperformed the (M + A) and (A + R) two-factor models in promoting the true peptide identity amongst all other candidates in each spectrum. From the perspective of FSR, the performance gain was consistent with the F-Rank: the FSR by the three-factor model (M = 0.782, SD = 0.00601) was higher than FSR of two-factor α model (M = 0.781, SD = 0.00462) t(4) = 3.14, p = 0.0349 (Fig. 1C). Further, the model "training" was found to be robust in that the optimal α and β values were stable across test sets (Supplementary Table 1).

We concluded that adding in RT prediction as a factor can further improve PSM candidate selection and we fixed the α and β values of the model to 0.71 and 0.08, respectively. The resulting model was termed "MARS" as abbreviation for the integration of **M**HC-prediction (M-score), **A**LC (A-score) and **R**etention-time prediction (R-score) for the improved **S**election of de novo sequence candidates.

## MARS improves MHC-associated peptide identification sensitivity across HLA-A and HLA-B alleles

To substantiate our findings, we applied MARS to synthetic HLA-peptide standard data for sensitivity evaluation. We hypothesized that the improvement of the re-ranking of de novo HLA-peptide identification by the MARS model is robust in the sense that it can be applied to different datasets with fixed factor coefficients. The only requirement is a transfer learning scheme needed to calibrate the DeepRTplus model to accommodate the discrepancy introduced by the LC system setups and gradient conditions between different datasets.

We utilized a large dataset published by Wilhelm, Zolg et al., of ~169,000 synthetic HLA class I peptides acquired by LC-MS[12]. Out of this dataset, we selected the 116 most frequent HLA alleles[33], comprising 35 HLA-A, 59 HLA-B, and 22 HLA-C alleles (Supplementary Table 2). We here included spectra from the 3xHCD method only, which was reported the best performing in the original study. Each peptide's HLA binding information was extracted from IEDB[34,35]. We further applied a filter to exclude peptide-HLA associations with NetMHCpan rank score > 10, which we deemed likely inaccurate assignments. This resulted in a collection of 28,921 MS2 spectra from 24,195 unique peptide sequences, of which the majority of sequences (26,456 spectra/22,327 peptide sequences; 92.3%) were listed in the Peaks DeNovo result candidate list (hence forward referred to as "recoverable") (Supplementary Table 2).

We selected sequence-RT pairs from high confidence peptide candidates with DeNovo A-score > 95 to calibrate the DeepRTplus model. MARS outperformed the Peaks DeNovo result, and the average F-ranks were significantly lower for MARS than the DeNovo result by paired Wilcoxon test (p < 2.2E-16, Fig. 2A). MARS FSR (M = 0.907, SD = 0.0344) was also significantly higher than the DeNovo FSR (M = 0.859, SD = 0.0744): t(115) = 8.4295, p = 9.9E-14 (Fig. 2B). MARS could again increase the number of identified peptide sequences over the DeNovo search alone (Fig. 2B), and overall improvement was 7.24%. We found that the peptide FSRs can be significantly improved by MARS for HLA-A and B-alleles, but not for HLA-C, for which the difference between DeNovo FSR and MARS FSR did not reach

significance, possibly due to a lower number of spectra included for HLA-C (Fig. 2C, Supplementary Table 2). When we further looked at the MARS performance for all individual alleles (Fig. 2D), and we noted that we could often observe a correction of the N-terminal two aa using MARS. This observation was most prominent for HLA-B*27:02 and A*25:01 (Fig. 2E). A variable MARS performance across different HLA alleles was observed, and HLA-B27 alleles stood out exhibiting the most striking improvement in peptide identifications using MARS (Fig. 2D). Interestingly, a disproportionately high percentage of HLA-B27 peptides was also observed amongst the 1,231 non-recoverable peptides that could not be recovered by the Peaks DeNovo candidate list (Fig. 2F), demonstrating a particularly low performance of the de novo sequencing for this allele.

## MARS identifies previously identified HLAp of non-canonical origins without integration of sample-specific RNA sequencing data

To further assess the identification performance of the MARS model, and in particular the performance in the non-canonical HLA peptide space, we utilized a published dataset from Chong et al. comprising nine LC-MS datasets for five melanoma cell lines and two sets of lung cancer and healthy control tissues spanning a wide range of HLA alleles (Supplementary Fig. 2)[6]. The authors identify and report 508 non-canonical, HLA-presented peptides mostly sourced from lncRNA genes, most of which are not present in the canonical human proteome. The authors achieved identification of these peptide sequences using an immunopeptidomics approach which carefully integrates RNA sequencing and ribosome profiling data for generation of personalized, sample-specific transcript assemblies. These were then used for in-silico translation into protein databases and mapping of non-canonical, HLA-presented sequences in the immunopeptidomics data using a standard database search.

Here, we wanted to understand whether MARS could recover these sequences without the integration of any of the additional RNA sequencing datasets used in the original publication. For this purpose, we selected the 2424 reported spectra, which gave rise to the 508 non-canonical sequence candidates in the original publication[6]. 2280 of these spectra contained the true (reported) sequence in de novo candidate list. We applied the MARS algorithm to the selected subset of spectra and established the FDR progression assuming that all 508 reported candidates were true identifications (Fig. 3A, Supplementary Table 3).

A clear separation of the number of peptides recovered with increasing FDR was observed when applying different MARS score thresholds in comparison to the DeNovo result alone (Fig. 3A). We therefore decided to define three main MARSscore thresholds, defining confidence regions with increasing global FDR: region 1: MARS score ≥ 95 (estimated FDR ≈ 27.6%), region 2: 95 > MARS score ≥ 90 (estimated FDR ≈ 41.2%) and region 3: 90 > MARS score ≥ 80 (estimated FDR > 41.2%) (Fig. 3A).

Overall, MARS was able to recover 317 of the 508 (62%) additional peptides reported by Chong et al. Notably, only 369 out of the 508 peptides were listed as a candidate in the Peaks DeNovo search results, hence the other sequences were "non-discoverable" by MARS. MARS therefore recovered 86% (317/369) of the possible "de novo-identifiable" space. Whilst we could not observe an allele bias in the group of peptides that were not discoverable by de novo sequencing, we observed generally lower mean ALC and MARS scores for the group of spectra that did not contain the true sequence candidate, indicating a general lower quality of these spectra (Supplementary Fig. 3).

With a stringent MARS score threshold of ≥ 95, and at an estimated 27.6% FDR, we were able to recover 255/369 (69%) of de novo-identifiable sequences, while the DeNovo search alone could recover only 165/369 (45%) (Fig. 3B), resembling an 1.6-fold improvement in FSR.

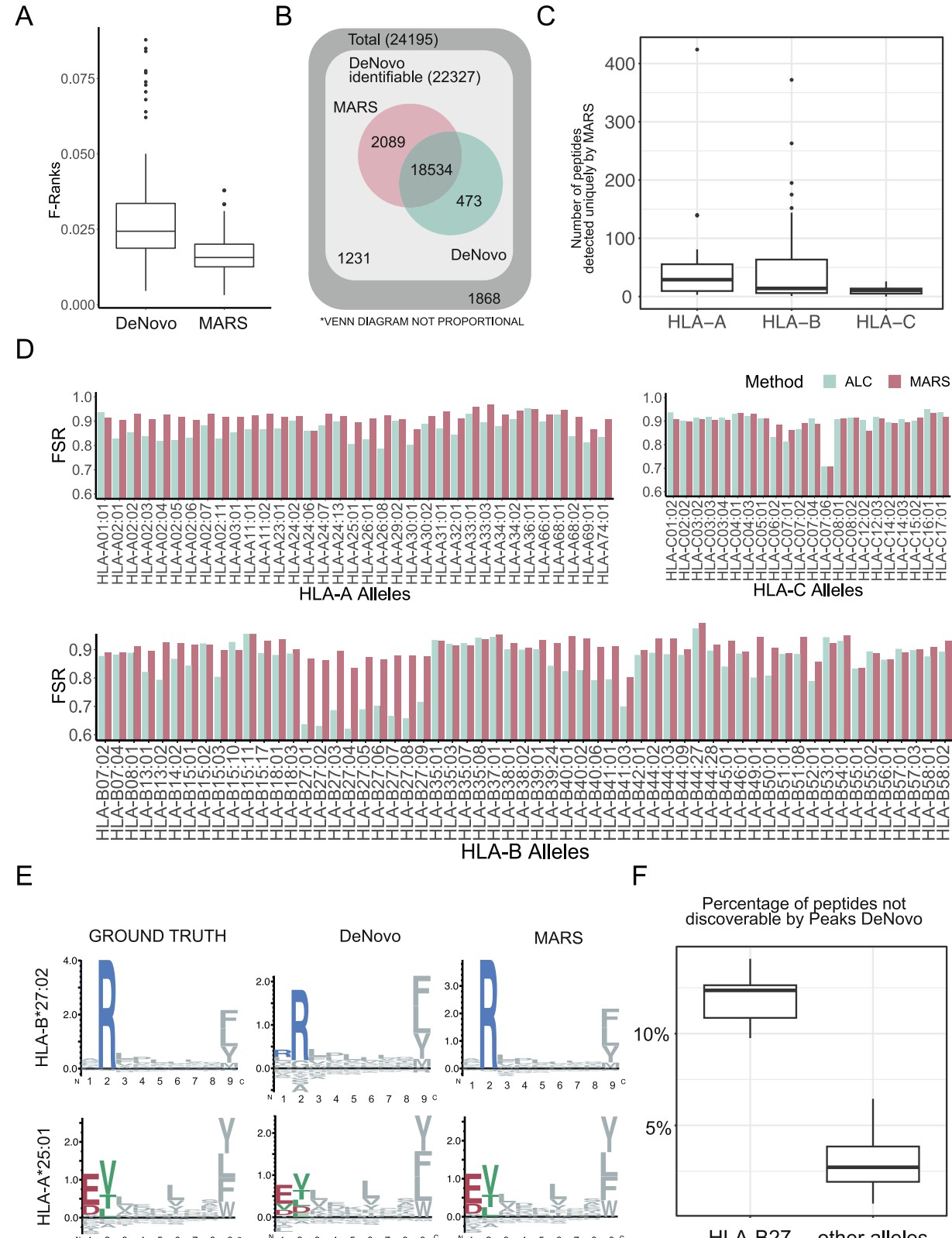

**Integration of a sequence mapping strategy for MARS peptide candidates confirms non-canonical HLA-peptide origin**

We then searched the complete Chong et al. datasets including all spectra against the SwissProt human protein database using the de novo-assisted DB search, in order to exclude spectra that were assigned to the canonical human proteome from our search. N-terminal acetylation and oxidation were included in the search parameters in order to match the analysis settings used in the original publication. We identified a total of 83,295 canonical peptides using the PEAKS database search and then applied the MARS model to the remaining spectra, returning an additional 380,893 unique peptide identifications.

We next aimed to assign the 380,893 MARS-identified sequences in the interrogated dataset to their respective likely gene origin. We

**Fig. 2 | MARS sensitivity evaluation in *Wilhelm* et al. large HLA peptide library datasets. A** The average F-rank distribution across all five partitions indicating first to third quartiles, including the median, minimal and maximal range by whiskers, and outliers are depicted as individual data points (**B**) Venn diagram depicting the number of peptide sequences identified by the DeNovo search (green) and MARS (red), respectively. The total number of spectra interrogated is shown in dark grey (24,195). Those spectra for which the correct peptide sequence was present in the 20 de novo candidates are reported as "de novo identifiable" in light grey (22,327). Out of these, 20,623 were correctly assigned by MARS, and 19,007 were correctly assigned by Peaks DeNovo alone, and 1231 neither Peaks nor MARS assigned correctly. **C** Box plots [as defined in (**A**)] indicating the number of peptide sequences uniquely identified by MARS for all HLA-A (35 alleles, 10,504 peptides), -B (59

alleles, 13,529 peptides), and -C (22 alleles, 3081 peptides) alleles combined. **D** Full sequence recall (FSR) for individual HLA alleles for the DeNovo result (green) and MARS (red). **E** Peptide motifs plot showing the amino acid frequency for all 9mers for indicated alleles for (i) the "ground truth" synthetic standard dataset, (ii) the Peaks "DeNovo" result, and (iii) "MARS" identifications. **F** Proportion of peptide sequences that were not recovered as a Peaks DeNovo candidate ("non-discoverable" peptides, 1868 total) which were allocated to HLA-B27 (378) in comparison to other alleles (1551). The boxes define the upper to lower quartiles and the median value is indicated within the box as horizontal line, while whiskers indicate the minimum and maximum values, respectively. Outliers are indicated individually as data points. Source data are provided as a supplementary file for Fig. 2A, C, D, and F.

therefore developed a stratified search approach to associate MARS-identified HLA-peptide to a wide range of possible origins across the human genome. We defined peptides mapping to (i) canonical origins mapping to known protein isoforms as "Human Protein", (ii) single aa substitutions as "1 aa substitution", and (iii) other locations in the human genome as "non-canonical" (Fig. 3C, and Methods section). For the 508 published sequences from Chong et al., 399 were defined as "non-canonical' using our criteria, 57 were "1 aa Substitutions" and 52 were designated as originating from "Human Protein" (Fig. 3D). Further to the 83,295 peptides that had initially been identified by the PEAKS database search, mapping of the 380,893 MARS identified sequences resulted in additional canonical assignment of 28,232 sequences mapped to SwissProt Isoforms, or other protein entries in Tremble and Ensemble. Another 26,386 MARS sequences mapped to canonical human proteins with exactly 1 aa substitution, which we considered as likely products of germline or tumour-specific mutations and a total of 15,966 peptides were linearly mapped to non-canonical sequences in the human genome (Fig. 3E, left panel). For 310,309 MARS peptides, we could not find an accurate match within the human genome with less than 1 aa distance ("unmatched", Fig. 3D, see Discussion and Supplementary Table 4).

From the 399 "non-canonical" peptides reported by Chong et al., 306 were "de novo-identifiable" as defined here in our approach, and 268/306 (87.6%) peptides were identified by MARS (Fig. 3E, right panel). In addition, with our search strategy, we found 16 peptides reported in our DB search, an additional 30 peptides matched to known human isoforms reported by SwissProt, Trembl and/or Ensemble, and 36 peptides to human proteins with 1 aa substitution (Fig. 3D, right panel).

The 268 peptides identified by both Chong *et al*. and MARS had an overall higher MARS score (mainly falling into regions 1 and 2) in comparison to the 15,966 additional non-canonical peptide candidates MARS recovered from the published datasets (Fig. 3F), indicating that these shared peptides were recovered with high confidence and that a higher MARS score cut-off can likely limit the FDR significantly.

## MARS extends the non-canonical peptide space by hundreds of peptides at an FDR < 14%

In order to determine the actual FDR for the identified MARS non-canonical peptide identifications experimentally, we selected 97 peptides that closely resembled the observed MARS score distribution and synthesized peptides for spectral validation and FDR evaluation. We also included 6 peptides identified by both Chong et al. and MARS, and 3 peptides matching to the human proteome as (likely positive) controls. We found 36 out of 97 sequences to be assigned correctly, resulting in an FDR of 14% (region 1, MARS score ≥ 95), 44% (region 2, 95 ≥ MARS score > 90), and 66% (region 3, 90 > MARS score ≥ 80) (Fig. 3G, H, Supplementary Data 1 and 2). All 9 positive control peptides were confirmed.

When considering only the 42 peptides that were matched to a genome origin (excluding all "unmatched" identifications), the FDR could further be reduced to 0% (region 1, MARS score ≥ 95), 37%

(region 2, > 90 MARS score ≥ 90), and 63% (region 3, > 90 MARS score ≥ 80) (Fig. 3H). This meant that all peptides with a MARS score of ≥ 95 that mapped to a genome origin were validated in this experiment.

We then proceeded to further stratify the non-canonical peptides using GENECODE "Biotype" definitions, including functional RNA genes and lncRNA genes, protein-coding regions, unannotated regions, and pseudogenes as detailed in the Method section (Fig. 3I). For this, we did not apply a hierarchical strategy that would consider certain origins more likely than others but argue that without sample specific genome/RNA level evidence, peptides with multiple possible origins cannot be faithfully assigned to a single source. Out of the 508 peptides originally published by Chong et al., 399 peptides are defined as "non-canonical" according to our categorization (Fig. 3D). MARS was able to expand the originally published, in our definition "non-canonical" peptide identifications from 399 originally published peptides in this category to 3,473 peptides from non-canonical origins at an FDR between 0 < FDR < 14% (Fig. 3I).

## MARS uniquely identifies additional non-canonical lncRNA-derived HLA peptide sequences in primary tumour tissue

To establish whether MARS could identify additional non-canonical, lncRNA-derived peptides in primary tumour tissue, we applied MARS to a set of in-house immunopeptidomics data from ten primary cervical tumour resections spanning 35 HLA alleles (Supplementary Table 5, Supplementary Fig. 2). Apart from 32,366 Peaks DeNovo-assisted DB search identifications, we could expand identifications using MARS by a further 77,129 assigned sequences. We mapped 6,551 additional peptides to human source proteins, 5828 peptides with a single aa substitution, and 3980 peptides to non-canonical origins (Fig. 4A). 60,770 peptides remained unmatched.

We then stratified the obtained non-canonical HLA-peptide according to the annotation of their origins as described in the last section. Peptides uniquely mapped to protein coding regions formed the largest subcategory, comprising 1150 HLA-peptides, while 811 peptides matched to unannotated regions. We found overall 1529 peptides associated with lncRNA origin, of which 414 were uniquely assigned to this origin and others overlapping with at least one other category (Fig. 4B).

To validate that the FDR of these identifications was as expected from the Chong et al. dataset for our defined confidence MARS score regions, we again selected a set of peptides for spectral validation. We chose 19 peptide candidates that had a MARS score of ≥ 90 (region 1 and 2), and we further prioritized peptides if more than one sequence supported the identification of an lncRNA gene.

We validated 11/19 spectra (42% FDR), which was in the range of the expected FDR for Region 1 and 2 (95 > MARS score ≥ 90). For Region 1 only (MARS score ≥ 95), we again yielded high accuracy in annotation with 6/7 correct assignments and an FDR of 12.5% (Fig. 4C, Table 1).

In summary, MARS identifies thousands of peptides in addition to peptides identified with standard DB search approaches at an accuracy

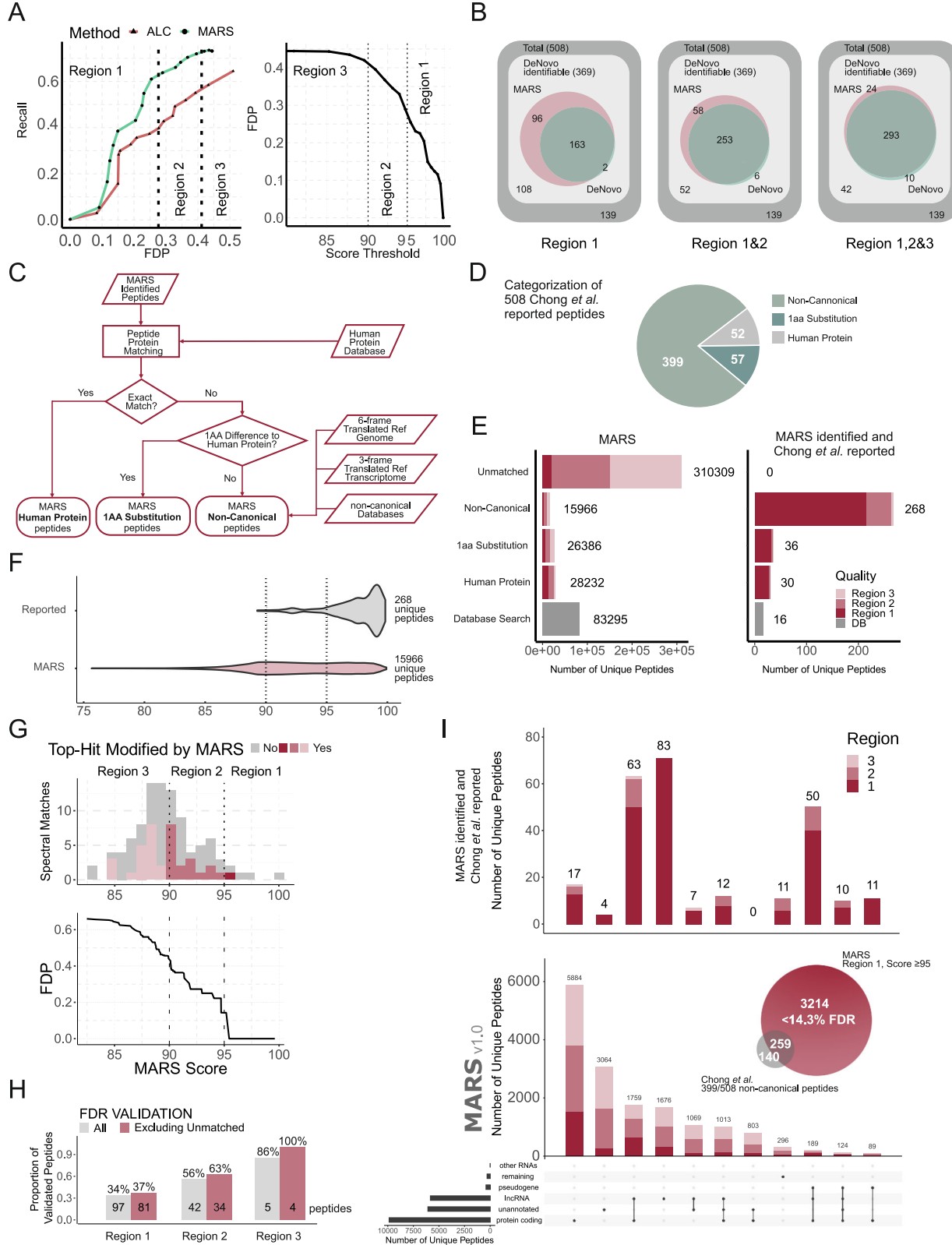

> 87% at a MARS score ≥ 95, as confirmed in this independent primary tumour dataset.

**Peptides uniquely identified by MARS are recognized by T cells**
Finally, we investigated whether the peptides that were uniquely identified by MARS in both the Chong et al. and cervical tumour datasets could be recognized by CD8⁺ T cells. We postulated that peptides that

were not observed in immunopeptidomics data from non-transformed tissues were more likely to be tumour-specific and to give an immune response in natural T cell repertoires. Hence, we reanalyzed the HLA Ligand Atlas dataset[36] using MARS and excluded peptides that had been identified in any healthy tissue interrogated within this database.

We shortlisted 2 peptides that were uniquely identified by MARS and assigned to lncRNA UNICODE classification which were

**Fig. 3 | Non-canonical peptide identification using MARS in a published dataset from Chong et al.[6]. A** Left panel: Sequence recall for the 369 de novo-identifiable lncRNA peptide sequences identified by Chong et al. across all samples plotted against the false discovery rate. A sequence recall of 1 equates to identification of all 369 peptides. The three regions are defined as follows: Region 1: FDR = 27.6%%, MARS score ≥ 95. Region 2: FDR = 41.2%, MARS score ≥ 90, Region 3: MARS score: ≥ 80. The right panel shows these defined confidence regions in relation to the MARS score. **B** Venn diagrams demonstrating the overlap of sequence identifications for each of the three defined regions from (**A**). A total of 508 spectra (dark grey space) were included here giving rise to the 508 peptide identifications reported by Chong et al. Those spectra for which the correct peptide sequence was present in the 20 de novo candidates are reported as "de novo identifiable" in light grey (369). 108, 52, and 42 spectra out of this group were not correctly identified by either engine in region 1,2, and 3, respectively. MARS and DeNovo identified peptide identifications are presented in red and green circles, respectively, for all three confidence regions. **C** Overview of MARS search engine and blast search strategy for identification of source origin. **D** Categorization of the 508 Chong et al. sequences using our sequence mapping strategy. **E** Left panel: Bar graph showing the categorization of the MARS hits as indicated. Right panel: Mutually identified peptides by MARS and Chong et al. from the 508 reported non-canonical identifications assigned to the MARS categories as indicated for the three confidence regions depicted in three shades of red (**F**) Score distribution of all MARS identified non-canonical peptides in comparison to those mutually identified by Chong et al. **G** Experimental validation of the FDR using a synthetic peptide standard. Histogram summarizing the peptide score distribution of the synthetic peptide standard. Shown are number of peptides identified by MARS only for indicated MARS score bins (upper panel), and FDP progression in relation to declining MARS score (lower panel) calculated on the spectral matching outcome. **H** Number and proportion of all confirmed peptides for all synthetic standard sequences and for the subset of peptides that were mapped to the human genome ("excluding unmatched"). **I** Upset plot of the MARS non-canonical peptide top 10 categories for all hits in region 1, 2, and 3 (lower panel), and mutual identifications as reported in Chong et al. (upper graph). The Venn diagram indicates the overall number of non-canonical sequences that were identified with MARS at a FDR ≤ 14.3% (red circle) out of the 399/508 sequences that were classified as non-canonical in our mapping strategy (grey circle). Source data are provided as a supplementary file for Fig. 3A, F, and G.

restricted to a frequent HLA allele (HLA-A*02:01 and HLA-A*03:01, respectively) (Table 2). The first peptide (SLFGVSERL) was restricted to HLA-A*02:01, and detected in patient ME275. It was mapped to an ORF in the long intergenic non-protein coding RNA 520 gene (LINC00520). The second peptide (VIITDVFQK) was restricted to HLA-A*03:01 and detected in a cervical tumour sample of patient 8. It was mapped to an ORF in the small nucleolar RNA host gene 6 (SNHG6).

Both peptides were confirmed with spectral matching (Supplementary Fig. 4), and fluorescent dextramers (Dex) were used to detect CD8+ T cells reactive to these peptides in peripheral blood mononuclear cells (PBMCs) from HLA-A2- and HLA-A3-positive healthy donors. Positive control multimers loaded with Flu peptides stained a fraction of CD8+ T cells that was in the expected frequency range[37,38]. It was reproducible across 3 PBMC replicates and was composed almost exclusively of effector/memory cells outside the CD45RA + CCR7+ gate identifying naïve T cells (Fig. 5A-C). As Dex+CD8+ T-cell fractions reactive to lncRNA-derived peptides were more subtle, the Dex+ gate was set in relation to a PBMC aliquot processed with the same staining panel with Dexs omitted (Fig. 5A, first column), and to the CD8− fractions of Dex-stained PBMC aliquots (Fig. 5A, third column)[38,39]. Data is summarized in Fig. 5B-C. Dex+ fractions were visual in the majority of donors, in a frequency range of 1-50/10⁶ CD8 + T cells expected for naïve T cells[38,39] for the HLA-A2-restricted SLFGVSERL peptide. Frequencies were, however, above this range for some donors for the HLA-A3-restricted peptide VIITDVFQK, which also displayed lower naïve fractions.

Collectively, these results show that the CD8 + T-cell repertoire comprises clonotypes capable of recognizing uniquely MARS-identified peptides and suggests the applicability for MARS in tumour-specific antigen discovery.

## Discussion

A main limitation of non-canonical antigen discovery is the requirement of prior knowledge of expected protein sequences translated in a given sample. We here describe how we can rescue identification of non-canonical lncRNA-derived HLA-associated peptides from LC-MS/MS data without the requirement for integration of RNA sequencing data or, in fact, any other sample-specific sequence information other than HLA type, using a discovery engine that improves candidate selection from de novo sequencing results.

Since in immunopeptidomics the lengths of peptides and positions of individual aa in the peptides are crucial for HLA binding and T-cell responses, we here used a study point-wise-full-sequence-recall (PSTR) to assess the development of MARS instead of improvements of single tag recall STR which assesses the percentage of individual amino acids being correctly identified[29,40].

We evaluated the increased sensitivity using an external dataset from Wilhelm et al.[12], and confirmed an overall increase in performance of MARS de novo candidate selection in comparison to the best scoring peptide as reported by Peaks DeNovo by over 7%. Interestingly, we saw disproportionately high performance of MARS to identify peptides originating from B27 alleles, whilst B27-derived peptides were also underrepresented in the Peaks DeNovo candidate list in the first place (Fig. 2D).

We did not observe a significant improvement for identification of HLA-peptides originating from HLA-C. Since peptide data was acquired in equimolar ratios for the HLA-peptide library datasets, this could not be due to lower expression of HLA-C in comparison to HLA-A and -B as generally observed across different datasets. Since NetMHCpan 4.1 performs equally well across all three loci, we conclude that most likely we did not evaluate a sufficient number of HLA-C peptide ligands to reach a significantly better performance.

We then proceeded to evaluate how many peptide identifications could be rescued from a list of published non-canonical HLA-peptides that had been generated using a proteogenomics approach. MARS was able to identify 259/399 (65%) of the reported sequences classified as non-canonical in our mapping strategy in the highest confidence interval (region1), highlighting the power of MARS to identify non-canonical antigens in the absence of transcriptomics and RiboSeq datasets.

One limitation remains the high FDR associated with de novo sequencing results. With MARS, the FDR could be further limited by assigning the peptide sequences to a possible origin in the human genome and transcriptome, resulting in a manageable FDR range < 14% for MARS scores ≥ 95, compared to generally anticipated FDR of 65% for de novo sequencing results[29]. Furthermore, considering only those sequences that mapped to a genome origin resulted in a substantial decrease of the FDR (to 0% in our experimental validation), demonstrating that MARS offers identification of non-canonical peptide sequences at low FDR, and can therefore expand the peptide identification beyond sophisticated proteogenomics approaches alone.

The origin of the large proportion of non-assigned peptide sequences identified by MARS remains unknown. Post-translationally modified peptides, peptides derived from proteasomal fusion events, and peptides of non-human origin are all potential candidates to explain this space, which requires further investigation. The observed higher proportion of sequences with lower MARS score within the unmatched peptides indicates a higher FDR in this subcategory, and further suggests that these spectra may not contain sufficient

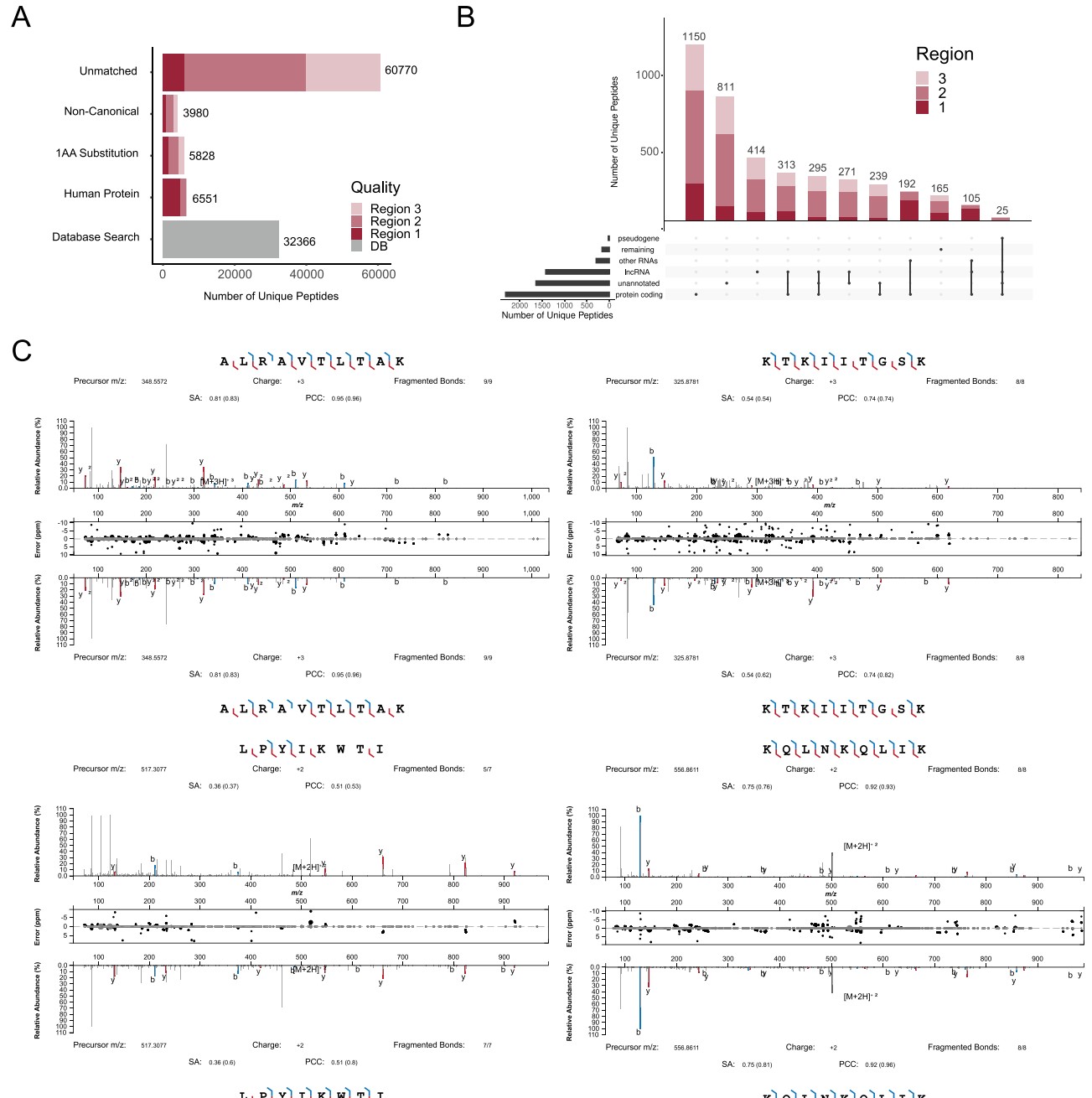

**Fig. 4 | Identification of a non-canonical HLAp in primary tumour tissue using MARS. A** Bar graph showing the categorization of the MARS hits as indicated. **B** Upset plot of the MARS non-canonical peptide category for all hits. **C** Spectral matching validation for indicated sequences identified in the primary tumour tissue. The upper spectra are the experimentally measured spectra, while the lower spectra are representing the acquired spectra from the synthetically synthesized counterpart molecule. SA: spectral angle; PCC: Princess Charlotte Coefficient as defined by the Universal Spectrum Explorer[60].

information for accurate sequence matching. This observation, however, also suggests that MARS performance could further be improved by measurement of higher quality spectra for each peptide ion species, i.e. by increased ion accumulation times.

We could not validate any cysteine-containing peptide assignments, confirming the known observed underrepresentation of cysteines in immunopeptidomics data[41,42], and highlighting the need for specific modification of this otherwise highly reactive aa.

We were able to shortlist additional lncRNA-derived peptide candidates from primary human tumour tissue. We shortlisted HLA-peptides if more than one sequence supported the identification of a specific lncRNA gene. We validated the sequence identifications with

spectral matching to their synthetic counterpart, and we validated the FDRs measured in our previous evaluation. Note that we did not set a threshold for the number of I and L in the peptide sequence, which could theoretically further reduce the FDR.

Finally, we identified and validated 11 peptides originating from lncRNA origins in human cervical cancer tissue. Peptide KVHVFLVKK originated from lncRNA HLA-F-AS1, which plays an oncogenic role in both colorectal cancer and breast cancer[43,44]. The identified peptides MTMSTILSK and MTMSTILSKK were derived from lncRNA lnckb.42318, also known as MIR4458HG (ENSG00000247516), which was reported as highly expressed lncRNA in 407 TCGA evaluated ovarian tumors[45]. It has further been reported that, together with

## Table 1 | lncRNA peptide validation in cervical cancer tissue

| PEPTIDE | MARS SCORE | HLA ALLELE | M-SCORE (ELR) | GENE LOCATION | LNC-RNA | ALT NAME | SAMPLE (CT) | SPECTRAL MATCH | FDP |
|---|---|---|---|---|---|---|---|---|---|
| **MTMSTILSKK** | 99.8 | HLA-A*11:01 | 0.12 | chr5:8460009-8460038 | lnckb.42318 | MIR4458HG | 4,8,9 | Yes | 0 |
| **KVHVFLVKK** | 99.7 | HLA-A*03:01 HLA-A*11:01 | 0.029 | chr6:29750586-29750560 | lnckb.45851 | HLA-F-AS1 | 1,8 | Yes | 0 |
| **MTMSTILSK** | 99.2 | HLA-A*11:01 | 0.008 | chr5:8460009-8460035 | lnckb.42318 | MIR4458HG | 8 | Yes | 0 |
| **KQLNKQLIK** | 98.3 | HLA-A*0301 | 0.053 | chr8:78259721-78259747 | lnckb.53745 | LOC105375911 | 4,9 | Yes | 0 |
| **ALRAVTLTAK** | 97.9 | HLA-A*03:01 | 0.099 | chr3:35175054-35175080 | lnckb.36918 | LOC101928135 | 8,10 | Yes | 0 |
| **VIFSGIRSL** | 95.4 | HLA-A*02:01 | 0.081 | chr13:94926124-94926150 | lnckb.14510 | LOC101927284 | 5 | Yes | 0 |
| *TELKAWKI* | 95.2 | *HLA-B*37:01* | 0.138 | *chr11:90693569-90693592* | *lnckb.9710* | *DISC1FP1* | 3 | No | 0.13 |
| **IPHGEIPDTSA** | 94.9 | HLA-B*56:01 | 0.064 | chr2:172323392-172323424 | lnckb.31346 | LOC107985960 | 5 | Yes | 0.13 |
| *VLLSIFIEHL* | 94.2 | *HLA-A*02:01* | 1.15 | *chr13:94845116-94845145* | *lnckb.14510* | *LOC101927284* | 7 | No | 0.18 |
| **KTKIITGSK** | 94.1 | HLA-A*03:01 | 0.192 | chrX:75675106-75675132 | lnckb.58303 | LOC107985664 | 8 | Yes | 0.18 |
| **LPYIKWTI** | 93.6 | HLA-B*51:01 | 0.076 | chr5:164805831-164805808 | lnckb.44609 | CTC-340A15.2 | 7 | Yes | 0.18 |
| *IIYFCLHKI* | 93.4 | *HLA-A*02:01* | 0.629 | *chr5:164989217-164989191* | *lnckb.44609* | *CTC-340A15.2* | 4,5 | No | 0.25 |
| *SLSILSLKV* | 92.9 | *HLA-B*13:02* | 0.164 | *chrX:75543586-75543560* | *lnckb.58303* | *LOC107985664* | 1 | No | 0.31 |
| *LPNFARII* | 92.6 | *HLA-B*07:02* | 0.29 | *chr2:172280278-172280255* | *lnckb.31346* | *LOC107985960* | 4,8 | No | 0.33 |
| *RQAPALRHL* | 92.5 | HLA-C*06:02 | 0.133 | chr13:94909331-94909357 | lnckb.14510 | LOC101927284 | 9 | Yes | 0.33 |
| *IISASKVIL* | 92.3 | *HLA-A*02:01* | 0.617 | *chr11:90809963-90809989* | *lnckb.9710* | *DISC1FP1* | 7,9 | No | 0.35 |
| **VTIFQNRVK** | 92.0 | HLA-A*11:01 | 0.575 | chr8:78413273-78413247 | lnckb.53745 | LOC105375911 | 8 | Yes | 0.35 |
| *IRNVKIYLI* | 91.2 | *HLA-C*06:02* | 0.09 | *chr6:29733441:29733437* | *lnckb.45851* | *HLA-F-AS1* | 10 | No | 0.39 |
| *LSSTLNKQI* | 90.1 | *HLA-C*02:02* | 1.143 | *chr5:164743628-164743602* | *lnckb.44609* | *CTC-340A15.2* | 1 | No | 0.42 |

Bold: Confirmed by spectral matching to synthetic peptide.
Italic: Not confirmed by spectral matching to synthetic peptide.
Underlined sequences: MARS identification not reported by Peaks DeNovo as the first candidate (MARS unique identification).

**Table 2 | Tumour-specific MARS-unique peptides (not reported by Peaks DeNovo as the first candidate) originating from lncRNAs identified in Melanoma (ME275) and Cervical Cancer (CT8) patient tissue**

| PEPTIDE | MARS SCORE | HLA ALLELE | M-SCORE (ELR) | GENE LOCATION | LNC-RNA | DESCRIPTION | SAMPLE | TUMOUR TYPE |
|---|---|---|---|---|---|---|---|---|
| VIITDVFQK | 98.04 | HLA-A*03:01 | 0.181 | chr8:66920561-66926398 | SNHG6 | small nucleolar RNA host gene 6 | CT8 | Cervical Cancer |
| SLFGVSERL | 95.63/95.22 | HLA-A*02:01 | 0.039 | chr14:55781132-55796731 | LINC00520 | long intergenic non-protein coding RNA 520 | ME275 | Melanoma |

CYTOR, and MAPTAS1, MIR4458HG was an independent prognostic factor in breast cancer patients (https://doi.org/10.21203/rs.2.24062/v1, preprint), with high expression associated with lower hazard ratios.

We finally selected one lncRNA-derived peptide from both the Chong et al. dataset, and our cervical cancer dataset, that derived from a lncRNA gene not observed in healthy tissue in the HLA atlas dataset[36] in order to understand whether we can detect T cell recognition in naive T cell repertoires. Both peptides were not correctly assigned by the Peaks DeNovo algorithm and were identified by MARS only.

SNHG6, which was here identified as presented by a cervical tumour, is a well-studied lncRNA and has been identified as expressed in multiple different tumour types[46–48]. and high SNHG6 expression is correlated with tumour progression and poor prognosis[49,50]. LINC00520, which was here sequences as presented in Melanoma cells, was first discovered in breast cancer[51], and has since been indicated as highly upregulated in cancer cells and has recently been suggested as a diagnostic and prognostic biomarker in cancer[52]. We were able to detect dextramer-positive T cells at levels similar to flu-antigen specificities, suggesting that both lncRNA-derived antigens can be recognized by T cells in human naive T cell repertoires.

We conclude that MARS is a suitable platform for non-canonical antigen discovery in cancer immunopeptidomics datasets in the absence of high-quality RNA sequencing data. We further highlight that the increases and unbiased results reported provide an exciting opportunity to expand non-canonical cancer antigen discovery to post-translationally modified peptides, peptides derived from proteasomal fusion events, and non-human sources, i.e. the microbiome.

## Methods

### Human tissue material

Tissue samples were obtained from Tissue Solutions Ltd.. Ethical approval was granted by the Central University Research Ethics Committee (CUREC) of the University of Oxford under reference R68126/RE001.

### Cell lines

All cell lines were mycoplasma-negative and cultured at 37 °C in the presence of 5% $CO_2$. The 721.221 HLA-class I deficient cell line that only expresses residual amounts of C*01:01[53] transfected with CD4 (CD4.221) and selected HLA class I alleles (A*01:01, A*02:01, A*03:01, A*11:01, B*08:01, B*44:02, B*57:03, C*03:03 C*03:04) was kindly provided by Prof. Masafumi Takaguchi, Kumamoto University, Japan. Cells were grown in RPMI 1640 medium (Thermo Fisher) containing 10% fetal bovine serum (FBS), 2 mM l-glutamine, 100 U/mL penicillin, and 100 μg/mL streptomycin (R10). R10 medium was supplemented with 0.15 mg/mL hygromycin B (Thermo Fisher) to maintain HLA-I expression as previously described[53,54].

### Baseline de novo A-score Model, or one-factor model

Peaks DeNovo sequencing algorithm was performed using Peaks X (Bioinformatics Solutions) on each MS2 spectrum in the studied data to produce peptide-spectrum matches (PSMs). The quality of PSMs are measured by the average local confidence (ALC, or A-score), which ranged in [0,100], with 100 being the best PSM quality. We selected PSMS with ALC in range [50, 100] to ensure reasonable input quality. The top-scoring peptide of each spectrum was used as its identity. For synthetic peptide validation datasets, we used 20 candidates per spectrum as spectra were high in quality. For HLA datasets, 100 peptide sequence candidates were considered per spectrum. No training was required for this one-factor model.

### Integration of MHC binding affinity ranking factor (M): two-factor (M + A) α model

To obtain numerical characterization of HLA-binding affinities, The PSM candidates were input into NetMHCpan 4.1, and the output eluted

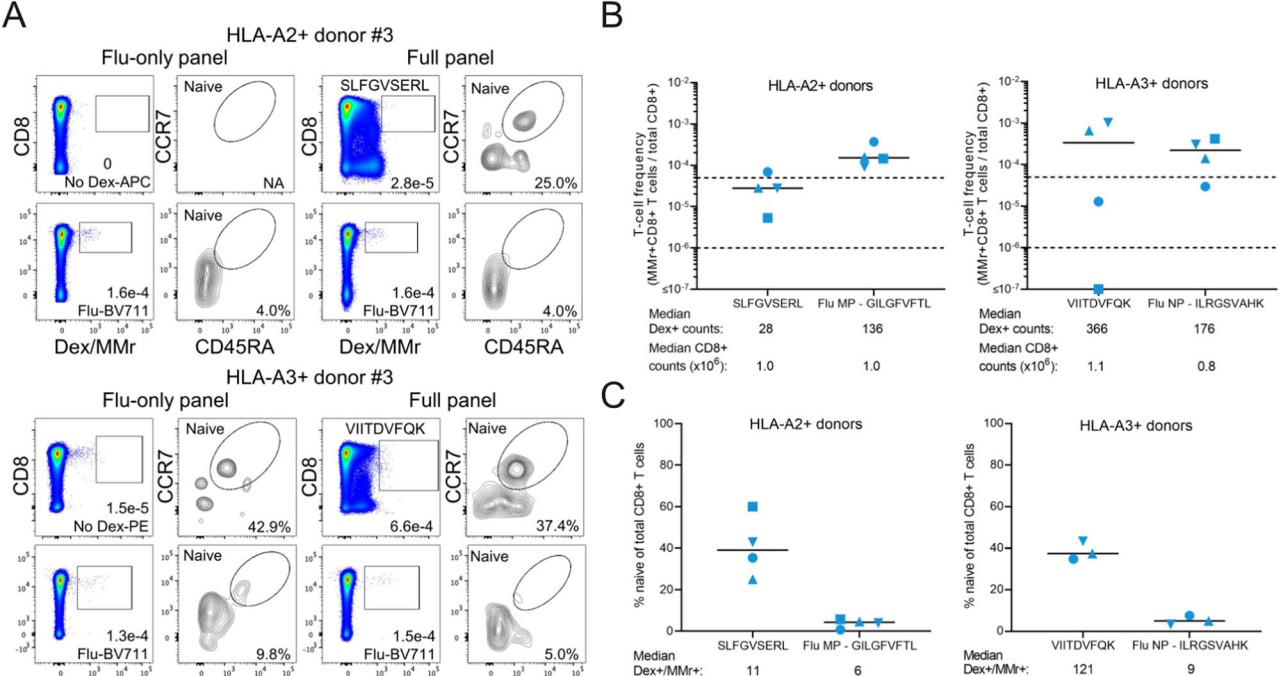

**Fig. 5 | Detection of circulating CD8 + T cells recognizing peptides uniquely identified by MARS. A** Staining of magnetically CD8-enriched PBMCs from a representative HLA-A2+ donor #3, and HLA-A3+ donor #3. Cells were stained with a BV711-labeled multimer loaded with Flu MP$_{58-66}$ peptide (MMr), alone (first two columns, Flu-only panel)) or in combination with APC-labelled HLA-A2 and PE-labelled HLA-A3 dextramers (Dex) loaded with the indicated peptide (third and fourth columns, full panel). Each dot plot is gated on viable CD3+ cells and indicates the frequency of Dex+ or MMr+ cells out of total CD8 + T cells. Contour plots on the right of each dot plot indicate the percent naïve (CD45RA + CCR7 + ) fraction out of

the corresponding Dex+ or MMr+ fraction. **B** Frequency of Dex+CD8+ cells out of total CD8 + T cells for the 4 HLA-A2+ (left panel) and HLA-A3+ (right panel) donors studied. Horizontal dotted lines indicate the expected T-cell frequency range of naïve CD8 + T cells. Bars indicate median values, with median Dex+CD8+ and total CD8+ counts indicated below each distribution. **C** Percent naïve cells out of total Dex+CD8+ cells for the same 4 HLA-A2+ and HLA-A3+ donors. Bars indicate median values; values were excluded for one HLA-A3+ donor (#2, square symbol) for whom < 5 Dex+ cells were counted. Median naïve Dex+CD8+ counts are indicated below each distribution.

ligand percentile rank (M-score) predictions were used to resemble the likelihood of MHC presentation of the corresponding peptides. The M-score predictions were considered robust across different biological samples and MHC alleles. The output M-score ranged between [0, 100], with the smaller ranks having higher likelihood of HLA presentation. We only selected PSMs with M-scores in the range [0,10] to exclude non-binding peptides. By incorporating M-scores into PSM candidates selection, a two-factor α model can be derived as: $(1-\alpha)$ $A + \alpha(100-M)$ which is a positive affine combination of the de novo ALC score (A-score) and predicted MHC binding rank score (M-score). The parameter α is in the range of [0,1], by the property of affine combination, the model outputs range is kept within [0,100] to ensure factors are normalized and of comparable scale.

### Integration of the RT factor (R), two-factor (A + R) β model and three-factor MARS model

We used an adapted version of DeepRTplus, which is a special type of deep neural network model termed capsule neural networks (Caps-Nets), to predict the RT of input peptides. Before the trained DeepRTplus model can be used for predicting into a new set of peptides, it needs to be recalibrated for each chromatographic setup, since peptides" RT observations are known to be chromatography dependent. We used the sequence-RT pairs from de novo-identified peptides with ALC ≥ 95 as a small but reliable set of identified peptides to re-calibrate the pre-trained DeepRTplus model, forming a transfer learning strategy to increase prediction accuracy.

To develop the two-factor (A + R) β model and the three-factor M + A + R model by incorporating the difference in peptide chromatographic RT out of the DeepRTplus output as the input, firstly the

absolute difference between observed and predicted RT: $\Delta T = |T_{predicted} - T_{observed}|$ was computed. Secondly, T was computed as the $\Delta T$ s normalised to the interval of [0,100] to match the range of A- and M-scores: $R = 100(\Delta T_{max} - \Delta T)/(\Delta T_{max} - \Delta T_{min})$. By this normalisation, R was transformed to a reciprocal score range, with high values being better than low values. Thirdly, to take both M-score and R-score into selecting de novo PSM candidates, a three-factor model was defined as: $(1-\alpha-\beta)A + \alpha(100-M) + \beta R$.

### Performance evaluation: five-fold cross-validation and F-rank score

For the evaluation of performance in our definition of the two-factor and three-factor models, we used five-fold cross-validation. Here, the data was shuffled randomly, and split into 5 random subsets of equal size. Model training was performed on four out of five groups and evaluated on the one remaining subset (test set). This test was repeated such that each subset was tested as a test set once (five times). Scores were calculated as the mean out of the five-fold validation exercise.

Fractional rank scores (F-ranks) of a correct peptide sequence for each spectrum is defined as the rank of this sequence in a list of all sequence candidates in descending order by score and divided by the total number of candidates of that spectrum. For peptides with tie scores, an average rank was used as the numerator. Therefore, the F-rank measures peptide identification performance improvement/deterioration at individual spectrum level. The reduction of mean F-rank of all correct peptides by multi-factor model outputs over the baseline de novo model was to suggest the promotion of the correct peptides. Differences in average F rank scores were evaluated with a paired T test.

**Performance evaluation: full-Sequence recall (FSR).** FSR is defined as the ratio between the total number of correctly identified peptides and the total number of peptides in the given dataset. It measures the peptide identification sensitivity at the spectra ensemble level. For each factor model, the top-scoring PSM of each MS2 spectrum was used as the identified peptide sequence, whereas the database peptide sequence of the same MS2 spectrum was considered the correct peptide sequence. A peptide was considered correctly identified if the peptide predicted by the given model was identical (up to I/L indistinguishability) to the database peptide. Differences in FSR were evaluated with a paired T test.

### False discovery proportion (FDP) calculation

We defined the peptide sequence level FDP as the number of correct sequence candidates (true positive, TP) devided by the total number of observed sequences (TP plus false positives, FP): FDP = 1-TP/(FP + TP).

For estimation of the FDP for MARS in the Chong et al. dataset, we selected the 2424 reported spectra, which gave rise to the 508 non-canonical sequence candidates in the original publication[6]. 2280 of these spectra contained the TP in *de-novo* candidate list and was used for FDP evaluation as depicted in Fig. 3A.

### Strategy for MARS candidate sequence origin annotation and database curation

We coarsely stratified the peptide sequence origins (encoding nucleotide sequence in the human genome) into three categories: "human protein", "single amino acid substitution", and "non-canonical". Besides, we put all sequences that were not strongly associated with either of these three categories into another "unmatched" category. Seqkit (Shen et al. 2016) was employed for locating MARS peptides in the curated databases. For these searches, we excluded all database search against SwissProt-assigned peptides with a score cutoff of -10$lg$P > 20 and searching the remaining de novo candidates as shortlisted by the MARS identification module.

Firstly, we searched Uniprot and Ensembl (only human protein entries) with isoforms as the "human proteome". MARS peptides matching exactly to human proteome were considered "Human Proteome" originated HLAp, matching human protein with 1 aa difference were considered "1aa substitution". These two categories of peptides were considered of canonical origin and excluded from the non-canonical category. To further examine the possible alternative origins of the peptides of non-canonical origin, we used GENCODE v40 six-frame translated reference genome and three-frame translated reference transcriptome to form the protein database. All annotations were combined into Genecode "Biotypes" as follows: "Protein coding": protein_coding; "lncRNA": lncRNA; "other RNAs": miRNA, misc_RNA, Mt_rRNA, Mt_tRNA, ribozyme, rRNA, rRNA_pseudogene, scaRNA, scRNA, snoRNA, snRNA, sRNA, vault_RNA; "pseudogene": all biotypes including "pseudogene" (polymorphic, processed, transcribed_processed, transcribed_unitary, transcribed_unprocessed, translated_processed, translated_unprocessed, unitary, unprocessed); "IG": IG_C_gene, IG_C_pseudogene, IG_D_gene, IG_J_gene, IG_J_pseudogene, IG_pseudogene, IG_V_gene, IG_V_pseudogene; "TR": TR_C_gene, TR_D_gene, TR_J_gene, TR_J_pseudogene, TR_V_gene, TR_V_pseudogened and "TEC": TEC. We integrated the following databases for lncRNA mapping: LNCipedia[55], lncRNAKB[56], and added a separate "TE" category using annotations from Dfam[57], and an Erv database[58]. We report our annotations in Upset plots to include multiple assignments.

### HLA-peptide purification from primary cervical tumour tissue

1 mL Protein A-Sepharose beads (GE Healthcare) were washed in 50 mM borate, 50 mM KCl (pH 8.0) solution and incubated with 2 mg of pan-HLA-I antibody (W6/32, ATCC HB-95) for 1 h at 4 °C. The beads were washed with 0.2 M triethanolamine (pH 8.2), and the bound antibody was cross-linked by incubation with 40 mM dimethyl pimelimidate dihydrochloride (DMP) (Sigma) (pH 8.3) for 1 h at room temperature. Ice-cold 0.2 M Tris buffer (pH 8.0) was added to the mixture to stop the reaction. Unbound antibody was washed off the column by washing with 0.1 M citrate (pH 3.0), and the column was equilibrated in 50 mM Tris (pH 8.0) for further use. Fresh-frozen tissue resections (0.6–0.9 g) were lysed by using 10 mL lysis buffer (0.5% IGEPAL 630, 150 mM NaCl, 50 mM Tris, pH 8.0, supplemented with protease inhibitor cocktail (Roche)), and homogenized using a bead-beater (Bertin Instruments). The lysate was centrifuged at 300 g for 10 min to remove nuclei and then at 15,000 g for 60 min to pellet other insoluble material. 1 mL W6/32 cross-linked to protein A-Sepharose beads (GE) was added to cleared lysates for 1 h, and beads were washed with 50 mM Tris buffer (pH 8.0) containing first 150 mM NaCl, then 450 mM NaCl, and next no salt. HLA-peptide complexes were eluted by using 5 mL 10% acetic acid and dried. After immunoprecipitation, peptide-HLA complexes were resuspended in 120 μL loading buffer (0.1% trifluoroacetic acid (TFA), 1% acetonitrile (ACN) in water). Samples were fractioned by reverse-phase (RP) high-performance liquid chromatography (HPLC) using an Ultimate 300 HPLC system (Thermo Scientific) supplemented with a 4.6- by 50-mm Pro-Swift RP-1S column (Thermo Scientific). Samples were loaded onto the column and eluted using a 10 min linear gradient from 3% to 30% ACN in 0.1% TFA at a flow rate of 500 nL/min, and elution was monitored by light absorbance at 280 nm. Fractions were collected in 1-min intervals. Alternate fractions were combined in two final fractions (odd and even).

### LC-MS acquisition

All synthetic peptides were obtained from Genscript as a library service and crude purity. We measured the peptide libraries and HLA-peptides by LC-MS on an Ultimate 3000 RSLCnano System coupled with an Q Exactive™ HF-X Hybrid Quadrupole-Orbitrap™ Mass Spectrometer (Thermo Scientific). Peptides were loaded onto the analytical column (PepMap C18 column, 2 μm particle size, 75 μm × 50 cm; Thermo Scientific) and eluted in a 60 min linear gradient from 3% to 25% ACN in 1% DMSO/0.1% formic acid at a flow rate of 250 nl/min. Peptides were introduced to the mass spectrometer using an EasySpray source at 2000 V and 45˚C, and the transfer tube temperature was set to 305˚C. Mass spectrometry (MS) detection was performed with a resolution of 120,000 for full MS (320-1600 m/z scan range) and AGC target of 300,000. A full-MS1 scan (120,000 resolution, 60 ms accumulation time, AGC 3×10$^6$) was followed by 20 data-dependent MS2 scans (60,000 resolution, 120 ms accumulation time, AGC 5×10$^5$), with an isolation width of 1.6 m/z and normalized HCD energy of 25%.

### LC-MS datasets and data analysis

We utilized the following datasets: Data for the C1866 cell line (A*01:01, B*08:01, B*44:02, C*05:01, and C*07:01) and for single allele transfectant cell lines is partially available in PXD015489[53]. Data for the Jurkat cell line (A*03:01, B*07:02, B*35:03, C*04:01, C07:02) is available in PXD011723[59]. Wilhelm et al. peptide standard datasets are available at PXD021013[12], and Chong et al. datasets are available at PXD013649[6].

Peaks X (Bioinformatics Solutions) was used to search against a database containing either the synthetic standard sequences (Wilhelm et al. dataset), or against the SwissProt-reviewed human database for all other searches. For the database search, we used no enzyme specificity (setting: unspecific). The precursor mass error range was set within 5 ppm. For the de novo sequencing, the precursor mass error range was set within 4.5 ppm, and 20 ppm for the 11 samples originating from patient T1185 in the Bassani-Sternberg dataset. Only oxidation of methionine was selected as variable modification, unless otherwise indicated in the main text.

### Spectral matching

Spectral comparisons were performed manually, with the help of the Universal Spectrum Explorer (USE)[60]. If a synthetic peptide was not

detected by LC-MS measurement, we considered the peptide sequence identification as "false", as it is equally unlikely that the identical peptide molecule was successfully ionized and detected during the experiment.

## Trinity de-novo RNA assembly

Raw RNA sequencing reads had adaptors removed and low-quality bases trimmed using Trim_Galore v0.6.2 (http://www.bioinformatics.babraham.ac.uk/projects/trim_galore) and quality control metrics assessed using FastQC v0.11.9 (http://www.bioinformatics.babraham.ac.uk/projects/fastqc). A custom STAR (v2.7.3a)[61] genome reference was created utilising UCSC hg38 genome sequence. Reads were then mapped to this custom genome reference using STAR with two passes, sorted and indexed by Samtools v1.10[62]. Duplicates were marked using GATK v4.1.7.0[63]. Unmapped reads from STAR were used as inputs for Trinity v2.11.0[64] for de novo transcript assembly.

## Shortlisting of tumour-specific peptides

Wie first shortlisted MARS peptides that were not identified by the DeNovo search with a MARS score ≥ 90. We excluded cysteine-containing peptides and peptides with more than 3 L/I residues. We then searched those candidates that mapped to a single unique ENSG gene location and shortlisted those peptides that were originating from ENSG genes that were not observed in any healthy tissue using the MARS reanalysed and reannotated data from the Human HLA Atlas[36]. Finally, we selected peptide candidates that were predicted to bind to HLA-A*02:01 and A*03:01 for selection of healthy donor PBMC for immunogenicity testing.

## Detection of MARS-identified peptide-reactive CD8+ T cells

APC-labelled HLA-A2 dextramers (Dex; Immudex) were loaded peptide SLFGVSERL; PE-labelled HLA-A3 Dex were loaded with peptide VIITDVFQK. BV711-labeled HLA-A2 multimers loaded with peptide Flu MP$_{58-66}$ (GILGFVFTL) and HLA-A3 multimers loaded with peptide Flu NP$_{265-273}$ (ILRGSVAHK) were produced as described[39], and used as respective controls.

Frozen-thawed PBMCs from 4 HLA-A2+ (2 males, 2 females, median age 40 years) and 4 HLA-A3+ donors (1 male, 3 females, median age 39 years) were magnetically depleted of CD8− cells (Stemcell Technologies) and stained with Dex and Flu multimers in the presence of 50 nM dasatinib, according to Immudex protocol. The following antibodies were subsequently added: CD45RA-FITC (RRID:AB_395879), CCR7-BV421 (RRID:AB_2728119), CD3-APC-H7 (RRID:AB_1645475), CD8-PE-Cy7 (RRID:AB_396852) and Live/Dead Aqua (ThermoFisher). After washing, cells were acquired on a BD LSRFortessa cytometer configured as described[39]. Data was analyzed using FlowJo 10.8 and GraphPad Prism 9. For each donor, Dex+ gates were set based on a PBMC aliquot stained only with Flu multimers and antibodies and on the CD8− fractions of Dex-stained PBMC aliquots.

## Reporting summary

Further information on research design is available in the Nature Portfolio Reporting Summary linked to this article.

## Data availability

Source data are provided for Figs. 1B, C, 2A, C, D, F, 3A, F, G as a Source Data file with the regarding subfigure title. The primary cervical tumour mass spectrometry data generated in this study is available under the identifier PXD046182. The publicly available mass spectrometry proteomics data utilized in this study is available at the ProteomeXchange Consortium (http://proteomecentral.proteomexchange.org) via the PRIDE partner repository[65], with the following identifiers: C1866 cell line and single allele transfectant cell lines: PXD015489. Data for the Jurkat cell line: PXD011723. Wilhelm et al. peptide standard datasets are available at PXD021013, and Chong et al. datasets are available at PXD013649. Source data are provided with this paper.

## Code availability

The code is propriety and is not accessible to commercial entities. Code will be provided to academic researchers upon request.

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

## Acknowledgements

We thank Prof. Masafumi Takiguchi, Kumamoto University, Japan for provision of CD4-expressing cell lines transfected with single HLA-I alleles. This work was funded by the NIH project #1R21AI153978-01 (LCE), The Leona M. and Harry B. Helmsley Charitable Trust project #1901-03689 and Agence Nationale de la Recherche project ANR-19-CE15-0014-01 (RM), the European Association for the Study of Diabetes FSD/JDRF/Lilly Programme on Type 1 Diabetes Research 2019 (NT, RM), Cancer Research UK Cancer Immunology project award C55884/A21045 (NT), and Cancer Research UK RadNet Centre Award C6078/A28736 (NT). ZZ was supported by a JDRF Postdoctoral Fellowship 3-PDF-2020-942-A-N. Some immunopeptidomic datasets employed for this study were generated as part of studies supported by Medical Research Council (MRC) programme grant MR/K012037, grants from the National Institutes of Health (UM1 AI 100645 and R01 AI 118549) and by AMED grant 17FK0410302H0003 for AIDS research (PB). The computational aspects of this research were supported by the Wellcome Trust Core Award Grant Number 203141/Z/16/Z and the NIHR Oxford BRC. The views expressed are those of the author(s) and not necessarily those of the NHS, the NIHR or the Department of Health. PB is a Jenner Institute Investigator.

## Author contributions

C.B. and Z.Z. contributed equally to this work. H.L. and N.T. conceptualized the work. N.T. supervised the project. H.L. led the coding and analysis of this work. M.N. supported the supervision of the model development, training and validation. P.B., L.C.E., R.M., M.N. and N.T. provided funding and resources. C.B. performed supporting analyses for model selection, training and validation. Z.Z. performed immunogenicity assays. X.P., I.W., A.T., R.P., W.P. performed experimental work and analyses in support of this manuscript. A.C., M.J.H., L.C.E., R.M. contributed with ideas, discussion and supervision to this project.

## Competing interests

NT is or has been a paid consultant Roche Pharma, Grey Wolf Therapeutics, T-Cypher Bio, Infinitopes, and Enara Bio. The remaining authors declare no competing interests.
