## [Peer Review File · Nature Communications]

MARS: Improved De Novo Peptide Candidate Selection for Non-Canonical Antigen Target Discovery in CancerREVIEWER COMMENTS

Reviewer #1 (Immunoproteomics) (Remarks to the Author):

General Comments:

This manuscript describes a de novo sequencing approach to interpret peptide sequences from high-quality MS/MS spectra in immunopeptidomics of HLA-associated peptide sequences (HLAp) that are unmatched after a preliminary search of the human reference proteome. These de novo sequences are then used for a sequence-based search of the “non-coding” portion of the transcriptome or genome to identify a potential protein-of-origin for each peptide amongst non-canonical sources like lncRNAs and pseudogenes. While these authors are not the first to employ this strategy, the primary contribution of this manuscript is to improve interpretation of the MS/MS spectra over the baseline de novo alone by applying machine learning to integrate two other independent properties of each peptide (i) HLA binding affinity predictions and (ii) differences between observed and predicted retention time (RT) of candidate peptide sequences. The approach is sound and if substantial improvements in clarity of presentation are made it would be suitable for the general audience of Nature Communications.

Because my greatest concern with this manuscript is the need to substantially improve clarity, I thought I would write this review a bit different than the usual list of concerns and suggestions. Instead, I’m going to try to convey the challenges I encountered trying to understand the manuscript as I read it. It would help to know that my primary expertise is generation and interpretation of MS/MS spectra in proteomics with interpretation by a variety of means including database search, spectral library matching, and de novo sequencing with immunopeptidomics being one area of focus. I read the manuscript in an order often used by scientists: abstract, last paragraph of introduction, each figure and legend, scan references to see if one’s own prior work is cited, then read results text with periodic detour to specific methods sections.

Major Comments:

1. Rationale: The abstract, introduction, and early results section of the current version of

the manuscript does not clearly articulate a rationale for the MARS approach. The reader is left pondering the questions: Why can't a de novo approach correctly interpret a sequence from an MS/MS spectrum? If one is going to add additional factors, why choose predicted binding affinity and predicted retention time? How can they be expected to contribute to improvement? Can one always use de novo sequencing, or only on the highest quality MS/MS?

I believe that rationale to be something like the following, please adapt as you see fit: Peptide fragmentation by MS/MS would ideally yield a complete set of ions derived from dissociation of each bond between adjacent amino acids to yield unambiguous sequence interpretation based on the mass differences between consecutive ions. However, individual fragment ions are often absent, and the resulting mass gap will support two or more amino acids, and the order of those amino acids in the sequence cannot be determined from the spectrum. To further complicate things, certain combinations of amino acids are isobaric (have the same mass). Hence, working with de novo sequence interpretations requires accommodating a limited degree of sequence ambiguity. In immunopeptidomics, the HLA class I bound peptides that are isolated will have predictable sequence motifs in position 2 and the C-terminus for the particular HLA alleles contributing to each sample. So applying MHC binding prediction to the best de novo candidate sequences for an MS/MS spectrum can be expected to select more favorable sequences from amongst the ambiguous possibilities, particularly the common de novo case of uncertain order of the first two amino acids. Chromatographic retention time (RT) of a peptide is an independent property of a peptide that could be used to further select amongst sequence candidates. RT prediction based on deep learning is now very accurate. While the predicted RT of peptides differing only in the order of 2 or 3 amino acids can not be expected to be resolved, RT prediction can be expected to help select amongst isobaric sequences of different amino acid combinations and a common de novo issue where most of a peptide sequence is reversed because the primary b or y ion series could not be unambiguously established. When incomplete fragmentation leads to N amino acids of ambiguous order and thus N! indistinguishable sequence permutations, MARS scoring aims to improve situations where the de novo sequence ambiguity ranges from 2-4 amino acids and identification must select amongst 2-24 sequence candidates/spectrum.

2. Figure 1 initial impressions prior to reading the text: What are F-ranks and full sequence recall? What are the 3 factors? Would it be interesting to include each 1 factor and 2 factor combination? I'm expecting that RT is a minor contributor because it can't be expected to resolve uncertainties of the relative order of di/tri peptides.

After reading text: Fig 1 is not referenced until paragraph 5 of Results, section 2! Consider reducing the preamble. Paragraph 2 could be moved to a supplemental section that includes some narrative accompanying the supplemental figures. In figure 1 the factors alpha and beta are not identified. The text does not clearly indicate that the two factor beta model is ALC + RT. I suspect that the acronym MARS was not adopted until late in the development/writing process. Now, clarity would be greatly enhanced if MARS was a central organizing principle of Fig 1 and the text. In the Fig 1 legend define M-A-R. In fig 1a place the items left to right in M-A-R order. Swap the HLA typing from right-side to left side. In Fig 1B –C and the text, replace the terms 1-factor, 2-factor alpha, 2-factor beta, and 3-factor with: A, M+A, A+R, M+A+R. In the fig 1 legend define F-rank and FSR or at least make it clear whether higher or lower values of each are better.

3. Figure 2 initial impressions prior to reading the text: ALC appears to have higher F-ranks, is a lower F-rank better? What are the 5 partitions? Why 20 candidates? Since the Venn is not proportioned there appears to be a lot of green that MARS misses. If proportioned it appears MARS gets only ~10% more(2087) than ALC alone. MARS misses 474. 1227 are missed entirely by PEAKS. Fascinating that PEAKS does very poorly on the B27 alleles which have an R in position 2 thus leading to spectra likely dominated by b-ions.

After reading text: The Fig 2B Venn diagram really should be replaced with one that portrays relative proportion. The text accompanying Fig 2D is very disappointing, it notes the alleles with aberrant performance, but provides little further insight. For B27 I suggest looking into the premise that Peaks scoring biases y-ion assignments over b-ion assignments. Thus when an allele has a position 2 anchor residue that is Arg, Lys, or His the spectra are generally likely to be b-ion dominated. Consequently, the PEAKS B27 top ALC scoring entries are more often likely to be y-ion dominated and thus reversed sequences with different end AA's. So MHC binding predictions and RT prediction should be more significant MARS

factors for the B27 alleles. Nothing in the text about the 1227 missed entirely by Peaks. Please address them. Perhaps start by examining their allele distribution.

4. Figure 3 initial impressions prior to reading the text: What are recall and FDP? Why are the region 1,2,3 boundaries set at 30% and 45%? Since the immunopeptidomics field is accustomed to operating in a world of <1% FDR, most of this figure isn't very meaningful and regions 2 and 3 don't merit consideration or inclusion in the figure at all.

After reading text: The figure has merit and steers the reader toward opting against de novo and instead toward database searching with databases that are informed by RNA-seq for noncanonical ORFs. Please clarify the Fig 3a y-axis. 1.0 would be recall of all 272 de novo identifiable sequences? The Fig 3B Venn diagrams really should be replaced with ones that portray relative proportion. For the Fig 3B white areas 96, 46, 38 peptides that were in the de novo results, but not MARS'd to the top, please say something about their characteristics. Also address the 236 gray area peptides that de novo could not find. What are some characteristics of those? These lead the reader to lean in favor of database searching.

5. Figure 4 initial impressions prior to reading the text: Again omit region 2 and 3 entirely. Legend describes panels C and D which are absent from the figure. Legend E) "Spectral matching validation for indicated sequences". Not clear what is being matched, synthetic peptides or perhaps Prosit predicted spectra? Are the spectra all from region 1 peptides?

After reading text: Region 2 and 3 peptides do not merit inclusion without better justification for why a very sloppy FDR should be tolerated. In the 4C legend clarify that the lower portion of each mirror plot is a synthetic peptide. Include the ALC scores for each of the upper spectra. Including the individual LC scores under each amino acid would help clarify the level of sequence ambiguity that results in a PEAKS result.

Minor Comments:

6. The term "affine combination" is used 5 times in the manuscript without an explanation of its meaning. After Googling the term, on behalf of other future readers I suggest using the

more recognizable English term linear rather than the Latin affine.

7. I was initially baffled by the name of the metric Full-Sequence Recall (FSR) which the methods define as the ratio between the total number of correctly identified peptides and the total number of peptides in the given dataset. The reason I was baffled is that the authors use it as a dataset level metric, while in de novo sequencing sequence recall is a long established spectrum-level performance metric that measures how much of a peptide's sequence can be unambiguously determined. So for de novo algorithm developments one presents the distribution of sequence recall for all spectra in a dataset.

8. Line 114 constituent "isoleucine (I) replaced by leucine (L), because Peaks DeNovo algorithm does not distinguish I from L." It's disingenuous to blame Peaks. More accurate to say that MS/MS fragmentation can not distinguish between the two amino acids Leu and Ile which have the same elemental composition and mass. Because Leu has 6 codons and Ile only 3, Leu occurs more frequently in proteins and is thus preferred when reducing sequences to only one of the two.

9. Line 133 "for each of the 100 sequence candidates in the PEAKS DeNovo list, and calculated for each peptide candidate an affine combination of the ALC and NetMHCpan predicted Rank (ELR) scores to produce a new composite score". Using the top 100 seems absurd, and conflicts with the number 20 in the Figure 2 legend. The depth of the Peaks result list to use should vary with the completeness of fragmentation of a peptide and could be better addressed for each spectrum by looking at the individual LC scores or for a significant drop in the average LC score amongst the candidate list. This concept could be discussed in the manuscript in the context of the MARS performance on the B27 alleles.

Reviewer #2 (Computational biology) (Remarks to the Author):

Overview

This manuscript reported an improved computational method, MARS, for enhancing de novo identification on HLA-associated peptide sequences for next generation immunotherapies in cancer. The authors highlighted two unique features in MARS, including

(1) integration of standard ALC (Average Local Confidence Score) with ELR (NetMHCpan MHC binding predicted Rank Score), DRT (Differences between Predicted and Observed Retention Time) for selection of de novo sequencing candidates. (2) independent pipeline without other types of data, such as RNA-seq or Ribo-seq data. Based on carefully selected training data, step-wise improvement of prediction models and performance validation with published mass spectrometry dataset, the authors demonstrated that MARS can achieve approximately double improvement in FSR (full sequence recall) evaluation on spectral assignments compared with standard de novo sequencing methodology and more than 85.7% assignment accuracy in non-canonical peptide space. In summary, MARS is proposed to be a more powerful platform to discover novel non-canonical peptides not only for HLA antigen discovery in cancer therapy but also for other potential application in proteomics. Immunotherapies are quite promising in cancer treatment. Toward to the success, one of the top priorities or challenges is to accurately identify the HLA-presented peptides, which can be used to develop efficient cancer vaccine, from LC-MS spectral data. Conventionally, peptide discovery heavily relies on the completeness of protein database, which disables the “database search” approach to discover novel, uncollected peptides. Despite de novo sequencing of peptides is currently available, but it may have unfavorable false positive rate. Meanwhile, RNA-seq data and Ribo-seq data are required to assist the peptide discovery, which more or less load additional work and complexity. More importantly, the authors point out considerable number of HLA-presented peptides are lowly expressed in RNA level, suggesting that RNA-seq data may be helpless in the discovery of such groups of peptides. Therefore, developing an independent pipeline for de novo peptide identification without any other information is quite desirable and significantly important in this field. Although MARS shows outperformance compared with standard approach, there are still several concerns about its performance, FDR, reliability and downstream utilization in antigen development (see below). In addition, some evaluation approaches may be less stringent and require to be improved.

Major concerns:

1. It can be expected to see an improved performance of MARS by integrating two additional factors (ELR and DRT) compared with ALC only, but the clearly improvement is not clear when it is compared with transcriptome-dependent approach. Indeed, MARS can

only recall 45% (227 in 508) lncRNA derived peptides from published data. In contrast, MARS generated a huge dataset, 333,011 unique peptides. This result leads to the concern that to recover very small number of true positives, MARS simultaneously brings up much larger number of potential false positives. The authors need to show more details to demonstrate the real performance of MARS in true positive discovery. In addition to lncRNA derived peptides, what about other types of non-canonical peptides? How about the discovery performance by MARS on these types?

2. For downstream application in cancer immunotherapies, it is the top priority to ensure that the identified HLA-presented antigen must be accurate. Transcriptome data is not only helpful in novel or sample-specific peptide discovery, but to provide additional layer of validation to ensure the accuracy of peptide identification. Moreover, without RNA-seq data, another validated approach may be required to confirm the prediction results. The authors should provide some discussion and insights on this issue. In addition, the author highlights the shortcoming of RNA-seq data on lowly expressed antigen, but there is no data available to show how MARS performs in such groups of antigens. For example, in the recovered group of peptides, how many peptides belong to lowly expressed assignment?

3. There is no detailed explanation about the reasons to incorporate ELR and DRT into the de novo prediction model. Are there any other factors we can consider in the future? Is there possible that the performance differences among HLA-A, HLA-B and HLA-C can be affected by the selected factors?

4. Does MARS need negative control dataset for training?

Minor concerns:

More details should be given for the training dataset. For example, how many spectra were selected for model training? How many spectra were filtered out during each step? What are the main results from training? There are too many methods description but rare demonstration on the results.

The three-factor model is just slightly better than two-factor model, do we really need the third factor? Can you explain the reason to choose retention time as a factor?

Have the authors evaluated the performance of MARS on canonical peptides?

How to evaluate $FDR < 14.3\%$ as an acceptable FDR?

Why use 1 AA substitution to distinct canonical and non-canonical peptides? What the

difference if using 2 AA?

Why did the authors only focus on lincRNA related non-canonical peptides? Is there any other genomic-feature derived peptide group?

Line233-234: I do not think it is an appropriate way to interpret the recovered rate.

Line255-256: All 1AA substitutions are products of mutations? Is possible that AA substitution may be sequencing errors?

Line307-309: According to the context, there are 1423 lincRNA associated peptide, 1150 protein coding, and 811 unmatched. Why did you write "protein coding formed the largest subcategory"?

Line318: how to get the accuracy above 85.7%?

Line 430: $ALC > 95$ or $ALC \geq 95$?

Can the authors describe the false discovery rate for currently available de novo sequencing methods? Which may be useful to show better performance of your method.

Writing and grammar mistakes:

There are many spelling or grammar mistakes through the entire manuscripts. Therefore, the next version must have intensive proofreading.

There are some examples:

Line 20-21: however, recent observations that are often low highlights...

Line 250 and 255: 1AA and 1 aa substitution should be consistent.

Line 322: is the beginning of DISCUSSION? Add section heading.

Line 442: Fractional Ranks (F-Ranks) repeat twice?

Line 513: it is equally unlikely that...

Reviewer #3 (Tumour epitopes) (Remarks to the Author):

In this manuscript, Liao and colleagues describe the potential of a novel method they developed called MARS (MHC binding prediction, Average Local Confidence Score, and Retention time integration for improved de novo candidate Selection) to identify and select HLA-presented non-canonical tumor epitopes. The topic is extremely timely considering our continued inability to consistently and accurately identify true tumor antigens for subsequent therapeutic targeting. Furthermore, the method exhibits much promise for

overcoming the existing challenges associated with identifying MS-derived epitopes, lack of correlation between transcriptional levels and HLA presentation, and limited ability to discover new tumor antigens. The manuscript is well written and the authors assess the potential of their method against a broad array of diverse HLA alleles. However, a few issues must be addressed before this work can be further considered for publication in Nature Communications.

Major issues

1. A monumental challenge facing in silico methods and identification of tumor antigens from mass spectrometry is the amount of noise which can emerge. Truly, the sole relevant attribute of a tumor antigen is to be both endogenously and immunogenic. As such, the authors should show that 1) a subset of the epitopes identified with MARS but not other methods are presented on the proposed HLAs via MHC stabilization assay and 2) that the antigens are immunogenic based on peptide stimulation, cytokine release, and/or assessment of antigen-specific cytotoxicity using PBMCs (healthy donors or patients) or TILs. I recognize this is a high ask but I believe it is a critical missing piece to this important story.

Minor issues

2. The differences in performance of MARS should be further elaborated upon to clarify what their potential causes may be. Is this solely due to lack of training and elution data for these specific alleles or are there other confounders?

3. Typographical error line 356: "parge proportion".

REPLY TO REVIEWERS' COMMENTS

NCOMMS-22-29737

"MARS: Improved De Novo Peptide Candidate Selection for Non-Canonical Antigen Target Discovery in Cancer"

We thank both reviewers for their helpful comments, which have improved our manuscript and raised its significance. Please note that during the review process we have found that the data for one patient from the published datasets in Chong *et al.* had a relatively high mass deviation. We reanalyzed all data with adjusted mass tolerances for this sample, and therefore the associated peptide identification numbers changed accordingly in our updated manuscript. To enhance clarity for the confidence regions, we have adjusted these to be defining clear score cut-offs, and, as a consequence, our adjusted FDR estimation is slightly improved.

Please find our detailed responses in green below.

Reviewer #1 (Immunoproteomics) (Remarks to the Author):

General Comments:

This manuscript describes a de novo sequencing approach to interpret peptide sequences from high-quality MS/MS spectra in immunopeptidomics of HLA-associated peptide sequences (HLAp) that are unmatched after a preliminary search of the human reference proteome. These de novo sequences are then used for a sequence-based search of the “non-coding” portion of the transcriptome or genome to identify a potential protein-of-origin for each peptide amongst non-canonical sources like lncRNAs and pseudogenes. While these authors are not the first to employ this strategy, the primary contribution of this manuscript is to improve interpretation of the MS/MS spectra over the baseline de novo alone by applying machine learning to integrate two other independent properties of each peptide (i) HLA binding affinity predictions and (ii) differences between observed and predicted retention time (RT) of candidate peptide sequences. The approach is sound and if substantial improvements in clarity of presentation are made it would be suitable for the general audience of Nature Communications.

Because my greatest concern with this manuscript is the need to substantially improve clarity, I thought I would write this review a bit different than the usual list of concerns and suggestions. Instead, I'm going to try to convey the challenges I encountered trying to understand the manuscript as I read it. It would help to know that my primary expertise is generation and interpretation of MS/MS spectra in proteomics with interpretation by a variety of means including database search, spectral library matching, and de novo sequencing with immunopeptidomics being one area of focus. I read the manuscript in an order often used by scientists: abstract, last paragraph of introduction, each figure and legend, scan references to see if one's own prior work is cited, then read results text with periodic detour to specific methods sections.

Reply: We thank the reviewer for their assessment and apologize that the manuscript was not clear. We improved the language throughout and have specifically addressed the comments of the reviewer as detailed below.

Major Comments:

1. Rationale: The abstract, introduction, and early results section of the current version of the manuscript does not clearly articulate a rationale for the MARS approach. The reader is left pondering the questions: Why can't a de novo approach correctly interpret a sequence from an MS/MS spectrum? If one is going to add additional factors, why choose predicted binding affinity and predicted retention time? How can they be expected to contribute to improvement? Can one always use de novo sequencing, or only on the highest quality MS/MS?

I believe that rationale to be something like the following, please adapt as you see fit:

Peptide fragmentation by MS/MS would ideally yield a complete set of ions derived from dissociation of each bond between adjacent amino acids to yield unambiguous sequence interpretation based on the mass differences between consecutive ions. However, individual fragment ions are often absent, and the resulting mass gap will support two or more amino acids, and the order of those amino acids in the sequence cannot be determined from the spectrum. To further complicate things, certain combinations of amino acids are isobaric (have the same mass). Hence, working with de novo sequence interpretations requires accommodating a limited degree of sequence ambiguity. In immunopeptidomics, the HLA class I bound peptides that are isolated will have predictable sequence motifs in position 2 and the C-terminus for the particular HLA alleles contributing to each sample. So applying MHC binding prediction to the best de novo candidate sequences for an MS/MS spectrum can be expected to select more favorable sequences from amongst the ambiguous possibilities, particularly the common de novo case of uncertain order of the first two amino acids. Chromatographic retention time (RT) of a peptide is an independent property of a peptide that could be used to further select amongst sequence candidates. RT prediction based on deep learning is now very accurate. While the predicted RT of peptides differing only in the order of 2 or 3 amino acids can not be expected to be resolved, RT prediction can be expected to help select amongst isobaric sequences of different amino acid combinations and a common de novo issue where most of a peptide sequence is reversed because the primary b or y ion series could not be unambiguously established. When incomplete fragmentation leads to N amino acids of ambiguous order and thus N! indistinguishable sequence permutations, MARS scoring aims to improve situations where the de novo sequence ambiguity ranges from 2-4 amino acids and identification must select amongst 2-24 sequence candidates/spectrum.

Reply: We sincerely thank the reviewer for pointing out that the abstract and introduction was not sufficiently explaining the rationale for our approach, and for taking the time to formulate these so clearly, it was very helpful to us. We noted that we did not sufficiently explain HLA sequence restriction - something we think about every day but of course is not obvious to the readership. We have now carefully reviewed the introduction, and integrated relevant sections on the basis of the reviewer's suggestions above, as highlighted in the resubmitted main manuscript.

Manuscript page 5/6: "De novo approaches have so far not often been integrated for peptide antigen discovery, since peptide fragmentation by MS/MS is never fully 'complete', and does

frequently lack consecutive ion series, particularly in MS/MS data in which the basic C-terminus and a strong y-ion series is not specifically generated by tryptic digestion. Gaps in the fragment ion series allow alternative sequence interpretation resulting in identical probability scores for isobaric amino acid combinations and sequence permutations.

This is specifically relevant for immunopeptidomics experiments, since the HLA gene locus is the most polymorphic in the human genome, and each individual HLA allele has the capacity to anchor peptides with specific amino acids in the main anchor positions, generally positions 2,3 and the C-terminus. Every person expresses up to six classical HLA proteins with such restricted sequence specificities. The resulting peptide repertoires that are purified and sequenced exhibit the respective sequence clusters formed by the HLA alleles present in each sample. Consequently, the quality of the measured spectra varies for each sequence subcluster, depending on the originating HLA allele sequence restriction.

Hence, we postulated that we could exploit our existing knowledge of the peptide sequence restriction defined by the HLA alleles present in the sample for 'filling the gaps' in incomplete fragment spectra, and ultimately achieve a more accurate sequence assignment. We had previously shown that HLA binding prediction can rescue low confidence identification in database approaches (MSRescue¹), and recently, machine learning has been applied in a personalized fashion in order to identify patient neoantigens².

Chromatographic retention time (RT) of a peptide is an independent property of a peptide that could be used to further select amongst sequence candidates. RT prediction based on deep learning is now very accurate. It has been shown difference between observed and predicted retention time (DeepRTplus³, Deep Rescore⁴) to database-search based peptide identification workflows can improve performance."

2. Figure 1 initial impressions prior to reading the text: What are F-ranks and full sequence recall? What are the 3 factors? Would it be interesting to include each 1 factor and 2 factor combination? I'm expecting that RT is a minor contributor because it can't be expect to resolve uncertainties of the relative order of di/tri peptides.

Reply: We have added more detailed explanations for F-ranks and FSR in the main text, and clarified the naming of the intermediate models as suggested. We realized that these are not generally used terms in the field. We had indeed evaluated both factors separately and have added this into the result section as suggested, and clarified outcomes in Figure 1 B/C. Indeed, as the reviewer suspected, RT was contributing much less dominantly than binding prediction.

Manuscript page 10: " We calculated the F-ranks which describe the rank of the true peptide sequence in the list of peptide sequence candidates ordered by score, divided by the total number of sequence candidates. The F-rank therefore provides a normalized performance measure for each spectrum and the smaller the F-rank scores, the better the sequence candidate scoring performs."

...

"To examine whether the improved F-rank led to higher identification sensitivity when top scoring peptide candidate per spectrum was taken as its identity, we employed another measurement,

the full sequence recall (FSR), which depicts the proportion of identified (top one scoring) peptides being correct."

After reading text: Fig 1 is not referenced until paragraph 5 of Results, section 2! Consider reducing the preamble. Paragraph 2 could be moved to a supplemental section that includes some narrative accompanying the supplemental figures. In figure 1 the factors alpha and beta are not identified. The text does not clearly indicate that the two factor beta model is ALC + RT. I suspect that the acronym MARS was not adopted until late in the development/writing process. Now, clarity would be greatly enhanced if MARS was a central organizing principle of Fig 1 and the text. In the Fig 1 legend define M-A-R. In fig 1a place the items left to right in M-A-R order. Swap the HLA typing from right-side to left side. In Fig 1B –C and the text, replace the terms 1-factor, 2-factor alpha, 2-factor beta, and 3-factor with: A, M+A, A+R, M+A+R. In the fig 1 legend define F-rank and FSR or at least make it clear whether higher or lower values of each are better.

Reply: We have implemented the reviewer's suggestions and updated the intermediate model naming in the updated version of the manuscript as follows: ALC with A-score; DRT with R-score, and ELR as M-score. The two-factor models are now referred to as (M+A) and (A+R), and the three-factor model is designated (M+A+R) for clarity. We appreciate that the first figure is referenced only in the second paragraph of the result section but were not sure whether the paper would benefit from a figure on the processing of the training data, which is explained in result section1. Therefore, we decided to keep the current references in place, and hope this is acceptable.

3. Figure 2 initial impressions prior to reading the text: ALC appears to have higher F-ranks, is a lower F-rank better? What are the 5 partitions? Why 20 candidates? Since the Venn is not proportioned there appears to be a lot of green that MARS misses. If proportioned it appears MARS gets only ~10% more(2087) than ALC alone. MARS misses 474. 1227 are missed entirely by PEAKS. Fascinating that PEAKS does very poorly on the B27 alleles which have an R in position 2 thus leading to spectra likely dominated by b-ions.

After reading text: The Fig 2B Venn diagram really should be replaced with one that portrays relative proportion. The text accompanying Fig 2D is very disappointing, it notes the alleles with aberrant performance, but provides little further insight. For B27 I suggest looking into the premise that Peaks scoring biases y-ion assignments over b –ion assignments. Thus when an allele has a position 2 anchor residue that is Arg, Lys, or His the spectra are generally likely to be b-ion dominated. Consequently, the PEAKS B27 top ALC scoring entries are more often likely to be y-ion dominated and thus reversed sequences with different end AA's. So MHC binding predictions and RT prediction should be more significant MARS factors for the B27 alleles. Nothing in the text about the 1227 missed entirely by Peaks. Please address them. Perhaps start by examining their allele distribution.

Reply: According to the reviewer's comments above, we have clarified in the main text what F-ranks stand for and how to interpret these (please see also previous response in point 2). Indeed, the smaller the F-rank scores, the better the sequence candidate scoring

performs, as the rank (the position) of the true peptide sequence is higher up in the list (its rank is lower).

Secondly, a more accurate description was added to explain the partitioning of the data in the relevant method section. Such partitioning of the data for cross-validation is a standard tool to evaluate performance of an ML model. We also added an explanation as to why 20 candidates were used here.

Page 24: For the evaluation of performance in our definition of the two factor and three factor models, we used five-fold cross validation. Here, the data was shuffled randomly, and split into 5 random subsets of equal size. Model training was performed on four out of five groups, and evaluated on the one remaining subset (test set). This test was repeated such that each subset was tested as a test set once (five times). Scores were calculated as the mean out of the five-fold validation exercise.

Page 22: For synthetic peptide validation datasets, we used 20 candidates per spectrum as spectra were high in quality. For HLA datasets, 100 peptide sequence candidates were considered per spectrum.

We agree with the reviewer that generally proportional Venn diagrams are beneficial, and we have amended Venn diagrams in figure 3 accordingly. However, in figure 2 the authors think that maintaining the non-proportionate diagram for clarity is preferred as the partitions of a proportionate Venn indicating uniquely identified sequences in MARS versus Peaks DeNovo would be too small due to the high overlap. We have indicated in the figure that the Venn diagram is non-proportional for clarity to the reader instead.

There are 1869 peptides missed by *de novo* sequencing entirely, and 1227 cannot be correctly identified as the best sequence candidate by ALC or MARS. We have now evaluated these group of spectra carefully.

Interestingly, we found that a disproportionately high number of "non-discoverable" peptides originate from HLA-B*27 assigned peptides (now added as Figure 2F). As the reviewer stated, we agree that this observation is likely due to a dominant b-ion series, and a bias introduced by the *de novo* algorithm which is trained on tryptic data, and we have added a statement to the discussion. We also looked at the peptide motifs between ALC only and MARS, and found that for a majority of the alleles, MARS corrected the position 1 and 2 in the peptide, which was pronounced in particular for A*25:01 and B*27:02 alleles (now displayed as examples in Figure 2E).

We have included these observations in the manuscript result section and added data to Figure 2.

*Page 13, Results: " When we further looked at the MARS performance for all individual alleles (Figure 2D), and we noted that we could often observe a correction of the N-terminal two aa using MARS. This observation was most prominent for HLA-B*27:02 and A*25:01 (Figure 2E). A variable performance across different HLA alleles was observed, and again HLA-B*27 alleles stood out exhibiting the most striking improvement in peptide identifications using MARS (Figure*

2D). Interestingly, a disproportionately high percentage of HLA-B27 peptides was also observed amongst the 1231 non-recoverable peptides that could not be recovered by the Peaks DeNovo candidate list (Figure 2F), demonstrating a particularly low performance of the de novo sequencing for this allele.

Page 18, Discussion: " Interestingly, we saw disproportionately high performance of MARS to identify peptides originating from B27 alleles, whilst B27-derived peptides were also underrepresented in the Peaks DeNovo candidate list in the first place (Figure 2D and Supplementary figure 2). B27 has the requirement of an arginine at position two within the peptide sequence. Hence, spectra are often dominated by a b-ion series, in contrast to tryptic peptides, that feature a basic amino acid at the C-terminus, resulting in a dominant y-ion series. Therefore, it might be conceivable that an intrinsic bias due to the specific training data used to train Peaks DeNovo algorithm is responsible for the observed discrepancies in performance."

Whilst we looked into the not identifiable sequences in the Chong *et al.* reanalysis and figure 3, we noted that one of the sample groups originating from one patient had much higher mass error than the other samples in the dataset. Reanalysis of these samples led to an overall higher performance of MARS, and greater sequence recovery from the set of published 508 peptide sequences. The figures and data reported now reflect these updated results.

We further interrogated the peptides that were still non-discoverable and could not find an HLA bias for this dataset. However, spectra were generally of lower quality as indicated by their average DeNovo ALC scores, and MARS scores. These data are now shown in Supplementary Figure 3, and explained in the manuscript:

Page 13: "Whilst we could not observe an allele bias in the group of peptides that were not discoverable by de novo sequencing, we observed generally lower mean ALC and MARS scores for the group of spectra that did not contain the true sequence candidate, indicating a general lower quality of these spectra (Supplementary Figure S3)."

4. Figure 3 initial impressions prior to reading the text: What are recall and FDP? Why are the region 1,2,3 boundaries set at 30% and 45%? Since the immunopeptidomics field is accustomed to operating in a world of <1% FDR, most of this figure isn't very meaningful and regions 2 and 3 don't merit consideration or inclusion in the figure at all.

After reading text: The figure has merit and steers the reader toward opting against de novo and instead toward database searching with databases that are informed by RNA-seq for noncanonical ORFs. Please clarify the Fig 3a y-axis. 1.0 would be recall of all 272 de novo identifiable sequences?

Reply: We want to firstly clarify that, if high-quality RNA sequencing data is available, the generation of databases using this information, and subsequent database searching can lead to the identification of high-confidence candidates. We have developed MARS for application to data where such information is not available, and for discovery beyond, i.e. what is missed with RNA sequence mapping.

Figure 3 was amended according to the reviewer's suggestions. We have chosen a MARS score cut-off rather than an FDR-defined cut-off to define our confidence regions for better clarity. We have carefully implemented a more detailed description of abbreviations used and cutoff's chosen and apologize for the lack of clarity. Please note that the number of recoverable peptides has increased sharply in the reanalysis using adjusted error tolerances (see comments above and below) for 11 runs from sample T1185.

Manuscript, Legend Figure 3: Left panel: "Sequence recall for the 369 de novo-identifiable IncRNA peptide sequences identified by Chong et al. across all samples plotted against the false discovery rate. A sequence recall of 1 equates to identification of all 369 peptides. The three regions are defined as follows: Region 1: FDR=27.6%, MARS score ≥ 95 . Region 2: FDR=41.2%, MARS score ≥ 90 , Region 3: MARS score ≥ 80 . The right panel shows these defined confidence regions in relation to the MARS score."

The Fig 3B Venn diagrams really should be replaced with ones that portray relative proportion.

Reply: We agree and have changed this accordingly in this figure.

For the Fig 3B white areas 96, 46, 38 peptides that were in the de novo results, but not MARS'd to the top, please say something about their characteristics. Also address the 236 gray area peptides that de novo could not find. What are some characteristics of those? These lead the reader to lean in favor of database searching.

The authors thank the reviewer for their important comment to interrogate the peptides that were non-discoverable. Indeed, this high proportion of peptides not identifiable with de novo sequencing was not according to our previous observations, and we found out that one of the samples, T1185 with 11 MS runs and raw files, exhibited a consistently high ppm error. This led to a very low identification rate in this sample, and most of the unidentifiable peptides were originating as a result of this high ppm deviation, which we had not noticed before. We therefore re-analysed the sample with a much higher (20 ppm) error tolerance. As a result, we found a highly improved performance of MARS, with 259/369 discoverable peptides identified. As a consequence of this change, Figure 3 had to be adjusted and numbers across the datasets have changed.

5. Figure 4 initial impressions prior to reading the text: Again omit region 2 and 3 entirely. Legend describes panels C and D which are absent from the figure. Legend E) "Spectral matching validation for indicated sequences". Not clear what is being matched, synthetic peptides or perhaps Prosit predicted spectra? Are the spectra all from region 1 peptides?

After reading text: Region 2 and 3 peptides do not merit inclusion without better justification for why a very sloppy FDR should be tolerated. In the 4C legend clarify that the lower portion of each mirror plot is a synthetic peptide. Include the ALC scores for each of the upper spectra. Including the individual LC scores under each amino acid would help clarify the level of sequence ambiguity that results in a PEAKS result.

Reply: The rationale for the separation into regions was to highlight that MARS can be applied flexibly, depending on the needs of the interrogation and anticipated FDR outcomes. We prefer to show all outcomes, so that the user can decide which FDR cut-off to choose depending on their validation approach.

Minor Comments:

6. The term “affine combination” is used 5 times in the manuscript without an explanation of its meaning. After Googling the term, on behalf of other future readers I suggest using the more recognizable English term linear rather than the Latin affine.

Reply: Affine combination refers to a special type of linear model where factor coefficients add up to 1. Please refer to Page 22 of "Convex Optimization"⁵ for reference, we have now included this in the manuscript for clarity, too.

7. I was initially baffled by the name of the metric Full-Sequence Recall (FSR) which the methods define as the ratio between the total number of correctly identified peptides and the total number of peptides in the given dataset. The reason I was baffled is that the authors use it as a dataset level metric, while in de novo sequencing sequence recall is a long established spectrum-level performance metric that measures how much of a peptide’s sequence can be unambiguously determined. So for de novo algorithm developments one presents the distribution of sequence recall for all spectra in a dataset.

Reply: We accept that a full-sequence recall can be used in different contexts with a different meaning. We hope that we could clarify our use of FSR, which we think we have unambiguously stated, is the proportion of those peptide sequences we can be fully recover (full sequence recall) from a given subset of sequences.

8. Line 114 constituent “isoleucine (I) replaced by leucine (L), because Peaks DeNovo algorithm does not distinguish I from L.” It’s disingenuous to blame Peaks. More accurate to say that MS/MS fragmentation can not distinguish between the two amino acids Leu and Ile which have the same elemental composition and mass. Because Leu has 6 codons and Ile only 3, Leu occurs more frequently in proteins and is thus preferred when reducing sequences to only one of the two.

Reply: Changed to."...because the here applied mass spectrometry approach cannot not distinguish I from L due to their identical atomic composition and mass."

9. Line 133 “for each of the 100 sequence candidates in the PEAKS DeNovo list, and calculated for each peptide candidate an affine combination of the ALC and NetMHCpan predicted Rank (ELR) scores to produce a new composite score”. Using the top 100 seems absurd, and conflicts with the number 20 in the Figure 2 legend. The depth of the Peaks result list to use should vary with the completeness of fragmentation of a peptide and could be better addressed for each spectrum by looking at the individual LC scores or for a significant drop in the average LC score

amongst the candidate list. This concept could be discussed in the manuscript in the context of the MARS performance on the B27 alleles.

Reply: Please also see comments above. Since most MARS rescored peptide sequences rank amongst the first 5 sequence candidates, the number of candidates used has very low weight on the outcome.

Reviewer #2 (Computational biology) (Remarks to the Author):

Overview

This manuscript reported an improved computational method, MARS, for enhancing de novo identification on HLA-associated peptide sequences for next generation immunotherapies in cancer. The authors highlighted two unique features in MARS, including (1) integration of standard ALC (Average Local Confidence Score) with ELR (NetMHCpan MHC binding predicted Rank Score), DRT (Differences between Predicted and Observed Retention Time) for selection of de novo sequencing candidates. (2) independent pipeline without other types of data, such as RNA-seq or Ribo-seq data. Based on carefully selected training data, step-wise improvement of prediction models and performance validation with published mass spectrometry dataset, the authors demonstrated that MARS can achieve approximately double improvement in FSR (full sequence recall) evaluation on spectral assignments compared with standard de novo sequencing methodology and more than 85.7% assignment accuracy in non-canonical peptide space.

In summary, MARS is proposed to be a more powerful platform to discover novel non-canonical peptides not only for HLA antigen discovery in cancer therapy but also for other potential application in proteomics.

Immunotherapies are quite promising in cancer treatment. Toward to the success, one of the top priorities or challenges is to accurately identify the HLA-presented peptides, which can be used to develop efficient cancer vaccine, from LC-MS spectral data. Conventionally, peptide discovery heavily relies on the completeness of protein database, which disables the “database search” approach to discover novel, uncollected peptides. Despite de novo sequencing of peptides is currently available, but it may have unfavorable false positive rate. Meanwhile, RNA-seq data and Ribo-seq data are required to assist the peptide discovery, which more or less load additional work and complexity. More importantly, the authors point out considerable number of HLA-presented peptides are lowly expressed in RNA level, suggesting that RNA-seq data may be helpless in the discovery of such groups of peptides. Therefore, developing an independent pipeline for de novo peptide identification without any other information is quite desirable and significantly important in this field. Although MARS shows outperformance compared with standard approach, there are still several concerns about its performance, FDR, reliability and downstream utilization in antigen development (see below). In addition, some evaluation approaches may be less stringent and require to be improved.

Major concerns:

1. It can be expected to see an improved performance of MARS by integrating two additional factors (ELR and DRT) compared with ALC only, but the clearly improvement is not clear when it is compared with transcriptome-dependent approach. Indeed, MARS can only recall 45% (227 in 508) lncRNA derived peptides from published data. In contrast, MARS generated a huge dataset, 333,011 unique peptides. This result leads to the concern that to recover very small number of true positives, MARS simultaneously brings up much larger number of potential false positives. The authors need to show more details to demonstrate the real performance of MARS in true positive discovery. In addition to lncRNA derived peptides, what about other types of non-canonical peptides? How about the discovery performance by MARS on these types?

Reply: The reviewer is pointing out the important point that MARS can identify many more sequences than those reported by the published proteogenomics approach. This is what we feel is the strength of MARS. However, we are not trying to say that MARS is performing better than an integrative proteogenomics approach, but that MARS is able to recover a large part of the carefully mapped sequences in Chong et al. in the absence of RNA/Ribo-Seq data. High quality RNA sequencing, and particular ribosomal profiling is not possible if only a small biopsy of tissue is available.

Our data demonstrates that we can not only recover more than half the peptides originally reported without integration of any additional RNA datasets. This outcome has further improved after our re-analysis as described above. Beyond this, we can significantly extend the discovery of non-canonical HLA-associate peptides at a controllable FDR between 0-14.3% - in our view adding substantial value to antigen discovery in primary tissues.

The identification of peptides by MARS is not limited to lncRNA sequencing and offers an unbiased view on the peptidome. The subsequent mapping of peptide sequences can be extended from the human genome to bacterial genomes and beyond.

2. For downstream application in cancer immunotherapies, it is the top priority to ensure that the identified HLA-presented antigen must be accurate. Transcriptome data is not only helpful in novel or sample-specific peptide discovery, but to provide additional layer of validation to ensure the accuracy of peptide identification. Moreover, without RNA-seq data, another validated approach may be required to confirm the prediction results. The authors should provide some discussion and insights on this issue. In addition, the author highlights the shortcoming of RNA-seq data on lowly expressed antigen, but there is no data available to show how MARS performs in such groups of antigens. For example, in the recovered group of peptides, how many peptides belong to lowly expressed assignment?

Reply: We thank the reviewer for their considerations but feel that the request to understand how lowly expressed RNA transcripts result in antigen presentation is beyond the scope of this study and has been shown by others, which is cited in our manuscript. However, we did show immunogenicity of peptides that were uniquely identified with MARS as a demonstration of their applicability for tumour immunotherapy, as requested by reviewer 3.

3. There is no detailed explanation about the reasons to incorporate ELR and DRT into the de novo prediction model. Are there any other factors we can consider in the future? Is there possible that the performance differences among HLA-A, HLA-B and HLA-C can be affected by the selected factors?

Reply: Indeed, as reviewer 1 has also pointed out, we apologize for not sufficiently explaining our rationale in the introduction and have now significantly extended this section in order to explain why we chose these two factors for our model in the manuscript.

Manuscript page 5/6: "De novo approaches have so far not often been integrated for peptide antigen discovery, due to the fact that peptide fragmentation by MS/MS is never fully 'complete', and does frequently lack consecutive ion series, particularly in MS/MS data in which the basic C-terminus and a strong y-ion series is not specifically generated by tryptic digestion. Gaps in the fragment ion series allow alternative sequence interpretation resulting in identical probability scores for isobaric amino acid combinations and sequence permutations.

This is specifically relevant for immunopeptidomics experiments, since the HLA gene locus is the most polymorphic in the human genome, and each individual HLA allele has the capacity to anchor peptides with specific amino acids in the main anchor positions, generally positions 2,3 and the C-terminus. Each individual expresses up to six classical HLA proteins with such restricted sequence specificities. The resulting peptide repertoires that are purified and sequenced exhibit the respective sequence clusters formed by the HLA alleles present in each sample. Consequently, the quality of the measured spectra varies for each sequence subcluster, depending on the originating HLA allele sequence restriction.

Hence, we postulated that we could exploit our existing knowledge of the peptide sequence restriction defined by the HLA alleles present in the sample for 'filling the gaps' in incomplete fragment spectra, and ultimately achieve a more accurate sequence assignment. We had previously shown that HLA binding prediction can rescue low confidence identification in database approaches (MSRescue¹), and recently, machine learning has been applied in a personalized fashion in order to identify patient neoantigens².

Chromatographic retention time (RT) of a peptide is an independent property of a peptide that could be used to further select amongst sequence candidates. RT prediction based on deep learning is now very accurate. It has been shown difference between observed and predicted retention time (DeepRTplus³, Deep Rescore⁴) to database-search based peptide identification workflows can improve performance."

4. Does MARS need negative control dataset for training?

Reply: The MARS method was trained to optimize the F-rank score, to allow improved ranking of the best sequence candidate in the de novo sequencing result. We used only high confidence hits for this optimization. In this scoring procedure, all peptides that are ranked first, but not identical to the database defined 'true' target sequence are assumed negatives. So given this, the MARS method does not require explicit negative data for training.

Minor concerns:

More details should be given for the training dataset. For example, how many spectra were selected for model training? How many spectra were filtered out during each step? What are the main results from training? There are too many methods description but rare demonstration on the results.

The three-factor model is just slightly better than two-factor model, do we really need the third factor? Can you explain the reason to choose retention time as a factor?

Reply: We have extended the introduction to explain why we chose to include RT prediction, please see our answer above. As this factor significantly improves the results ($p=4.3 \times 10E11$) as described in the result section, so there is no rationale to remove it.

Have the authors evaluated the performance of MARS on canonical peptides?

Reply: Yes, we used five-fold cross-validation for validation of MARS performance. All these peptides were canonical identifications.

How to evaluate $FDR < 14.3\%$ as an acceptable FDR?

Reply: The aim here is to understand how high the FDR is in the first instance. For antigen discovery, we feel that it is valuable to have 10 potential tumour-target candidates and knowing that 8 of them are likely true identifications that may be relevant for clinical targeting. Since the FDR is scalable with the MARS score, the threshold can be set by individual research based on how much validation they are able to do.

Why use 1 AA substitution to distinct canonical and non-canonical peptides? What the difference if using 2 AA?

Reply: The aim here was to consider frequent germline mutations, often observed 1 aa distance.

Why did the authors only focus on lincRNA related non-canonical peptides? Is there any other genomic-feature derived peptide group?

Reply: We focussed on lincRNAs here due to the direct comparison with the Chong *et al.* dataset in which mostly lincRNAs were identified. MARS is not limited to the identification of lincRNAs, as described in the manuscript.

Line233-234: I do not think it is an appropriate way to interpret the recovered rate.

We are reporting the number of peptides MARS can recover from those spectra, where the true sequence candidate is listed, and have termed the latter "*de novo* recoverable". We

are further now also reporting how many peptides out of the total reported peptides were identified by MARS. We hope this is now acceptable.

Line255-256: All 1AA substitutions are products of mutations? Is possible that AA substitution may be sequencing errors?

Reply: With these analyses we wanted to make sure to capture germline differences in the proteome of different individuals in comparison to the Swissprot annotated genome. It is not likely that these are sequencing errors per se, but of course some of these sequences will not be assigned correctly as estimated by our FDR analysis. It is to note that these AA substitutions are defined in comparison to a reference genome.

Line307-309: According to the context, there are 1423 lncRNA associated peptide, 1150 protein coding, and 811 unmatched. Why did you write "protein coding formed the largest subcategory"?

Reply: We corrected this in the manuscript. We cited the largest subcategory for which peptides were uniquely assigned, which was "protein coding". Whilst there were a higher number of lncRNA identifications, only 414 peptides were uniquely assigned to this origin, while other peptide sequences in this category had at least one other association with other subcategories.

Manuscript, page 16: "Peptides uniquely mapped to protein coding regions formed the largest subcategory, comprising of 1150 HLAp, while 811 peptides matched to unannotated regions. We found overall 1,529 peptides associated with lncRNA origin, of which 414 were uniquely assigned to this origin and others overlapping with at least one other category (Figure 4B)."

Line318: how to get the accuracy above 85.7%?

Reply: We validated the FDR by spectral matching using synthetic counterpart peptides as described in the manuscript and found the FDR to be between 0-14.3%. In fact, in the in-house validation cervical tumour dataset, we achieved an FDR of 12.5%. For lower FDR and higher accuracy, a higher MARS score threshold can be chosen. However, since we had validated all candidates with a MARS score of ≥ 95 in our FDR validation, we suggest to use this as a general cut-off, which will result in an FDR of up to ~14.3%.

Line 430: $ALC > 95$ or $ALC \geq 95$?

Reply: Corrected

Can the authors describe the false discovery rate for currently available de novo sequencing methods? Which may be useful to show better performance of your method.

Reply: Yes, we have cited this in the manuscript. The global FDR of de novo sequencing results is around 65%.

Manuscript, page 7, introduction: "... offering opportunity for improved shortlisting of accurate sequence identifications from the generally observed global 65% false discovery rate (FDR) in de novo sequence data⁶".

Manuscript, discussion, page 20: "With MARS, the FDR could be further limited by assigning the peptide sequences to a possible origin in the human genome and transcriptome, resulting in a manageable FDR range below 14.3% for MARS scores above 94.8, compared to generally anticipated FDR of 65% for de novo sequencing results."

Writing and grammar mistakes:

There are many spelling or grammar mistakes through the entire manuscripts. Therefore, the next version must have intensive proofreading.

There are some examples:

Line 20-21: however, recent observations that are often low highlights...

Changed

Line 250 and 255: 1AA and 1 aa substitution should be consistent.

Done

Line 322: is the beginning of DISCUSSION? Add section heading.

Inserted

Line 442: Fractional Ranks (F-Ranks) repeat twice?

Corrected

Line 513: it is equally unlikely that...

We could not find an error in this sentence

Reviewer #3 (Tumour epitopes) (Remarks to the Author):

In this manuscript, Liao and colleagues describe the potential of a novel method they developed called MARS (MHC binding prediction, Average Local Confidence Score, and Retention time integration for improved de novo candidate Selection) to identify and select HLA-presented non-canonical tumor epitopes. The topic is extremely timely considering our continued inability to consistently and accurately identify true tumor antigens for subsequent therapeutic targeting. Furthermore, the method exhibits much promise for overcoming the existing challenges associated with identifying MS-derived epitopes, lack of correlation between transcriptional levels and HLA presentation, and limited ability to discover new tumor antigens. The manuscript is well written and the authors assess the potential of their method against a broad array of diverse HLA alleles. However, a few issues must be addressed before this work can be further considered for publication in Nature Communications.

Reply: We thank the reviewer for their overall positive comments.

Major issues

1. A monumental challenge facing *in silico* methods and identification of tumor antigens from mass spectrometry is the amount of noise which can emerge. Truly, the sole relevant attribute of a tumor antigen is to be both endogenously and immunogenic. As such, the authors should show that 1) a subset of the epitopes identified with MARS but not other methods are presented on the proposed HLAs via MHC stabilization assay and 2) that the antigens are immunogenic based on peptide stimulation, cytokine release, and/or assessment of antigen-specific cytotoxicity using PBMCs (healthy donors or patients) or TILs. I recognize this is a high ask but I believe it is a critical missing piece to this important story.

Reply: We understand the viewpoint of the reviewer. Other bioinformatics studies published in Nature journals to evidently improve peptide identification⁷ do not include immunogenicity data, and we do not report clinical targets in our manuscript.

However, we agree that immunogenicity is crucial in order to demonstrate applicability of MARS for tumour antigen discovery. We have therefore now expanded the results, and included identification of two tumour-associated peptide sequences, that were uniquely identified by MARS, and not *de novo* sequencing alone. We demonstrate immunogenicity, and further show that the reactive T cells from healthy volunteers were stimulated from naive repertoires.

We hope that these data is sufficient to demonstrate the power and applicability of MARS for tumour antigen discovery.

We have not performed requested binding studies as we generated dextramers from the identified peptides, which were stable, and functional in our T cell assays as shown.

Minor issues

2. The differences in performance of MARS should be further elaborated upon to clarify what their potential causes may be. Is this solely due to lack of training and elution data for these specific alleles or are there other confounders?

Reply: This is a very good point that was raised by reviewer 1 as well. We have looked at the differences in performance and noted that MARS often corrects the assigned amino acid position for the first two amino acids in the peptide. This correction is due to the NetMHCpan algorithm correctly predicting the relevant anchor residue. These observations are now included in Figure 2 and the manuscript results.

3. Typographical error line 356: “parge proportion”.

Reply: We thank the reviewer for spotting this error, which is now corrected.

REFERENCES:

1. Andreatta, M. et al. MS-rescue: A Computational Pipeline to Increase the Quality and Yield of Immunopeptidomics Experiments. *Proteomics*, e1800357 (2018).
2. Tran, N.H. et al. Personalized deep learning of individual immunopeptidomes to identify neoantigens for cancer vaccines. *Nat Mach Intell* **2** (2020).
3. Ma, C. et al. Improved Peptide Retention Time Prediction in Liquid Chromatography through Deep Learning. *Anal Chem* **90**, 10881-10888 (2018).
4. Li, K., Jain, A., Malovannaya, A., Wen, B. & Zhang, B. DeepRescore: Leveraging Deep Learning to Improve Peptide Identification in Immunopeptidomics. *Proteomics* **20**, e1900334 (2020).
5. Boyd, S.P. & Vandenberghe, L. Convex optimization. (Cambridge University Press, Cambridge, UK ; New York; 2004).
6. Muth, T. & Renard, B.Y. Evaluating de novo sequencing in proteomics: already an accurate alternative to database-driven peptide identification? *Brief Bioinform* **19**, 954-970 (2018).
7. Wilhelm, M. et al. Deep learning boosts sensitivity of mass spectrometry-based immunopeptidomics. *Nat Commun* **12**, 3346 (2021).

REVIEWERS' COMMENTS

Reviewer #1 (Remarks to the Author):

For all of the issues that I previously commented on, that the authors addressed, the revised manuscript is greatly improved in clarity and readability. The addition of the fig 5 T-cell reactivity data is excellent. So this review will only touch on the item to which the authors did not respond. This should not require me to see a further revision, the journal editorial staff should be able to handle the author's response.

Figure 4 Legend E) "Spectral matching validation for indicated sequences". Not clear what is being matched, revise to clarify that they are matching spectra from synthetic peptides (lower) against the original experimental observation (upper). Please indicate for each peptide which region 1,2,3 it is from. Also give a brief explanation of the meaning of the labels SA (spectral angle?) and PCC (Princess Charlotte Coefficient?). By contrast the legend of Figure 5 is so clear and thorough it can be readily understood by someone unfamiliar with T cell assays. The accompanying text uses that terribly vague phrase "spectral matching" again. Please think of the non-specialist reader as you clarify the Fig 4E legend.

[EDITORIAL NOTE: Reviewer #2 was late so we asked reviewer #3 to mediate]

[EDITORIAL NOTE: name redacted], I've reviewed the comments and rebuttal. To me it seems like they were properly addressed but these are computational in nature and this is outside of my expertise. Not sure how you feel about this.

[EDITORIAL NOTE: Reviewer #1 also mediated these comments]:

I just re-read the rebuttal responses to reviewer 2's concerns. I understand the concerns and believe the authors have sufficiently addressed them.

[EDITORIAL NOTE: This process lengthened the review process and during the AIP decision reviewer #2 also submitted comments.]

Reviewer #2 (Remarks to the Author):

Overall, I still have some concerns about the FDR control for MARS. Compared with the previous de novo prediction method, MARS has shown significant improvement at higher FDR levels but similar performance at low FDR levels, such as 0.05 or 0.1. Please see the following questions:

Major Concerns:

1. In Line 296-297, there are 310,309 MARS peptides that cannot be matched to the human genome. Could you discuss the reasons for such a high "unmapping rate"? Are these peptides real or incorrectly predicted?
2. In Fig. 3A, the recall is quite low when setting the FDR at 0.05 or 0.1 for both methods (de novo and MARS). We can observe that after setting the FDR at 0.1, the recall has dramatically increased. Could you explain what factors cause this phenomenon? Why do these methods show such low recall at more stringent FDR levels?
3. From Line 352-354, the conclusion suggests that MARS can achieve >87% accuracy at a MARS score ≥ 95 for thousands of peptides identified by MARS. Could you please clarify how you calculate this figure of 87%?

Minor Issues:

- LINE 131: Why choose 13 as the maximum length, if the typical length is between 8-11 according to the previous studies?
- LINE 134: How many spectra pass the criteria to make up the training dataset? Please provide the exact number.
- Figure 2B should be clarified further. Several numbers in this Venn diagram are difficult to understand, such as 1231 and 1868. It would be helpful to specify which parts these numbers correspond to. Additionally, in the Figure legend, please indicate which part corresponds to the "20 de novo candidates" (LINE 685).
- Figure 2E: Labels are too small.

- Figure 2F: The y-axis label is missing.
- Figure 3B: Same suggestions as for Figure 2B.
- Figure 1, $-10\lg p \geq 20$, LINE 126, and LINE 130: Are $-10\lg P$ and $-10\log P$ the same or not? They should be consistent with each other.

Reviewer #3 (Remarks to the Author):

I appreciate the authors' responses to my critiques and have no further comments.

Alexandre Reuben

REVIEWERS' COMMENTS

Reviewer #1 (Remarks to the Author):

For all of the issues that I previously commented on, that the authors addressed, the revised manuscript is greatly improved in clarity and readability. The addition of the fig 5 T-cell reactivity data is excellent. So this review will only touch on the item to which the authors did not respond. This should not require me to see a further revision, the journal editorial staff should be able to handle the author's response.

We thank the reviewer for their time to review our manuscript carefully, and for all comments provided. We hope our final responses are acceptable to the editorial.

Figure 4 Legend E) "Spectral matching validation for indicated sequences". Not clear what is being matched, revise to clarify that they are matching spectra from synthetic peptides (lower) against the original experimental observation (upper). Please indicate for each peptide which region 1,2,3 it is from. Also give a brief explanation of the meaning of the labels SA (spectral angle?) and PCC (Princess Charlotte Coefficient?). By contrast the legend of Figure 5 is so clear and thorough it can be readily understood by someone unfamiliar with T cell assays. The accompanying text uses that terribly vague phrase "spectral matching" again. Please think of the non-specialist reader as you clarify the Fig 4E legend.

We have updated figure legend 4E and added the region for each peptide. Table 1 includes the MARS score for each peptide, and therefore indicates the confidence region for each molecule. We added missing abbreviations according to the reviewer's suggestion, and cited the tool used in the legend for more clarity.

[EDITORIAL NOTE: Reviewer #2 was late so we asked reviewer #3 to mediate]

[EDITORIAL NOTE: name redacted], I've reviewed the comments and rebuttal. To me it seems like they were properly addressed but these are computational in nature and this is outside of my expertise. Not sure how you feel about this.

[EDITORIAL NOTE: Reviewer #1 also mediated these comments]:

I just re-read the rebuttal responses to reviewer 2's concerns. I understand the concerns and believe the authors have sufficiently addressed them.

[EDITORIAL NOTE: This process lengthened the review process and during the AIP decision reviewer #2 also submitted comments.]

We thank the editor and both reviewers for helping with the additional evaluation and for trying to speed up the process!

Reviewer #2 (Remarks to the Author):

Overall, I still have some concerns about the FDR control for MARS. Compared with the previous de novo prediction method, MARS has shown significant improvement at higher

FDR levels but similar performance at low FDR levels, such as 0.05 or 0.1. Please see the following questions:

Major Concerns:

1. In Line 296-297, there are 310,309 MARS peptides that cannot be matched to the human genome. Could you discuss the reasons for such a high "unmapping rate"? Are these peptides real or incorrectly predicted?

Yes, indeed, the majority of spectra, as commonly observed in LC-MS2 experiments, remain unmatched. Mass spectrometry yield highly specific spectra for peptide identification, those spectra are of high specificity, but due to incomplete fragmentation, most of the spectra may not uniquely match to a peptide sequence. In a database search, the missing information comes from high quality protein database, and the remaining sequence can be "inferred" through this information. For our *de novo* sequencing approach, the matching to the human transcriptome and genome is helping to refine those sequence annotations that are correct. However, since the quality and completeness of the information in the peptide spectrum highly depends on the peptide's abundance in the sample, its capacity to ionise, its chemical modification, etc., the majority of spectra in an LC-MS2 experiment remain not sufficient in quality to allow a confident assignment of a single peptide sequence.

We have discussed the unmatched peptide space in the discussion: "*The origin of the large proportion of non-assigned peptide sequences identified by MARS remains unknown. Post-translationally modified peptides, peptides derived from proteasomal fusion events, and peptides of non-human origin are all potential candidates to explain this space, which requires further investigation. The observed higher proportion of sequences with lower MARS score within the unmatched peptides indicates a higher FDR in this subcategory, and further suggests that these spectra may not contain sufficient information for accurate sequence matching. This observation however also suggests that MARS performance could further be improved by measurement of higher quality spectra for each peptide ion species, i.e. by increased ion accumulation times.*"

2. In Fig. 3A, the recall is quite low when setting the FDR at 0.05 or 0.1 for both methods (de novo and MARS). We can observe that after setting the FDR at 0.1, the recall has dramatically increased. Could you explain what factors cause this phenomenon? Why do these methods show such low recall at more stringent FDR levels?

Yes, indeed the performance of both ALC and MARS are comparable at very low FDR, which is driven by the proportion of spectra from the most abundant ions species formed during acquisition. *De novo* sequencing alone will be able to assign the complete sequence with high accuracy in these spectra. MARS, however, can help if the spectrum is of lower quality, and missing information has to be inferred accordingly.

3. From Line 352-354, the conclusion suggests that MARS can achieve >87% accuracy at a MARS score ≥ 95 for thousands of peptides identified by MARS. Could you please clarify how you calculate this figure of 87%?

Since we have estimated a FDR of 12.5%, we reformulated this observation to an overall sequence accuracy of >87%.

Minor Issues:

- LINE 131: Why choose 13 as the maximum length, if the typical length is between 8-11 according to the previous studies?

There is a slight discordance in the literature, but it has been shown that slightly longer ligands than 11mers can bind to HLA. We did not want to exclude slightly longer peptides in our work. However, the number of peptides matching this longer length is minor.

- LINE 134: How many spectra pass the criteria to make up the training dataset? Please provide the exact number.

There are 118,177 spectra that pass the criteria to make up the training dataset. We have included this information in the manuscript.

- Figure 2B should be clarified further. Several numbers in this Venn diagram are difficult to understand, such as. It would be helpful to specify which parts these numbers correspond to. Additionally, in the Figure legend, please indicate which part corresponds to the "20 de novo candidates" (LINE 685).

We have added more information to the figure legend.

- Figure 2E: Labels are too small.

Corrected

- Figure 2F: The y-axis label is missing.

The title of the graph cites the y-axis units

- Figure 3B: Same suggestions as for Figure 2B.

We have added more information to the figure legend.

- Figure 1, $-10\lg p \geq 20$, LINE 126, and LINE 130: Are $-10\lg P$ and $-10\log P$ the same or not? They should be consistent with each other.

Corrected

We thank the reviewer for their thorough review, which have improved our manuscript significantly.

Reviewer #3 (Remarks to the Author):

I appreciate the authors' responses to my critiques and have no further comments.

Alexandre Reuben

We thank the reviewer for their helpful comments, which have enhanced the impact of our study.